# Targeting the ceramidase ACER3 attenuates cholestasis in mice by mitigating bile acid overload via unsaturated ceramide-mediated LXRβ signaling transduction

Leyi Liao [1,8], Ziying Liu[2,8], Lei Liu[3,8], Can Huang[1], Yiyi Li[4], Cungui Mao[5], Ruijuan Xu[5], Haiqing Liu[1], Cuiting Liu[6], Yonghong Peng[6], Tingying Lei[7], Hanbiao Liang[1], Sheng Yu[1], Jianping Qian[1], Xianqiu Wu[1], Biao Wang[1], Yixiong Lin[1], Jie Zhou[1], Qingping Li [1] ✉, Chuanjiang Li [1] ✉ & Kai Wang [1] ✉

Bile acid overload critically drives the pathogenesis of cholestatic liver injury (CLI). While ceramide metabolism has garnered increasing interest in liver research, the role of ceramides in CLI remains unclear. This study investigates the function of alkaline ceramidase 3 (ACER3)-catalyzed hydrolysis of unsaturated ceramides in CLI. Using clinical specimens, this work finds that *ACER3* expression is upregulated in the cholestatic liver and positively correlated with the severity of CLI in patients. *Acer3* ablation increases ceramide(d18:1/18:1) and attenuates bile duct ligation-induced CLI in female mice with reduced hepatic necrosis, inflammation, and fibrosis. However, it does not significantly impact CLI in male mice. Moreover, ceramide(d18:1/18:1) treatment attenuates CLI in wild-type female mice. Similarly, *ACER3* knockdown and ceramide(d18:1/18:1) treatment prevent lithocholic-acid-induced cell death in human-liver-derived HepG2 cells. Mechanistically, ceramide(d18:1/18:1) binds the ligand binding domain of the liver X receptor β, acting as an agonist to activate its transcriptional functions. This activation upregulates sulfotransferase 2A1-catalyzed bile acid sulfation, normalizes bile acid metabolism, and restores lipogenesis, thereby reducing bile acid overload in hepatocytes to attenuate CLI. Our findings uncover the role of ceramide(d18:1/18:1)-liver X receptor β signaling in mitigating bile acid overload in the cholestatic liver, offering mechanistic insights and suggesting therapeutic potential for targeting ACER3 and ceramide(d18:1/18:1) for CLI.

Cholestatic liver injury (CLI) contributes to the progression of various liver diseases, including viral hepatitis, biliary cholangitis, and cholelithiasis, leading to liver malignancy and liver failure[1]. CLI results from impaired bile formation or flow and accumulation of bile within the cholestatic liver[2]. Bile acids (BAs), the major component of bile, are key causative factors of CLI[3]. Disruption of BA excretion and metabolism lead to BA overload, which destroys cell membranes, disturbs lipid metabolism, triggers inflammation, and induces cell death in the cholestatic liver[2–4]. Significant efforts are still being made to understand the compensatory

mechanism against BA overload and identify therapeutic targets for CLI[5].

Nuclear receptors (NRs), such as ligand-dependent liver X receptors (LXRs) and farnesoid X receptor (FXR), are key transcriptional factors that maintain the metabolic homeostasis of BA and lipid in the liver[6]. Certain lipid derivatives act as endogenous ligands of NRs[7]. For instance, oxysterols (OS) bind to the hydrophobic pocket of LXRs and activate their transcriptional functions[8]. Sulfotransferase (SULT) expression controlled by NR is essential for countering BA overload[9]. SULTs transfer sulfated groups from 3′-phosphoadenosine-5′-phosphosulfate to BAs, forming hydrophilic BA-sulfates that reduce BA toxicity and promote their excretion from the kidney[10]. BA sulfation is suppressed in the cholestatic liver due to dysregulated SULT expression, which impairs BA detoxification and elimination to exaggerate BA overload[11]. NRs also regulate genes involved in lipogenesis, orchestrating lipid metabolism to maintain the lipid levels for physiological homeostasis[12]. However, cholestasis disrupts NR signaling transduction, resulting in impaired BA detoxification and lipid homeostasis[13].

Ceramides (CERs) are a class of bioactive sphingolipids that crucially regulate pathologic processes associated with CLI, including cell death, inflammation, and fibrogenesis[14]. Dysregulation of CER metabolism has garnered increasing research interest in CLI. Hepatic CER levels were found to be increased through the degradation of complex sphingolipids in the mouse liver with CLI[15,16]. Additionally, elevated plasma levels of saturated-long-chain CERs have been linked to the severity of intrahepatic cholestasis in pregnant women[17]. The function of CERs in regulating CLI is not fully elucidated. Early research classified CERs as a broad category of detrimental lipids due to their pro-death effects[16–18]. The CER metabolite sphingosine-1-phosphate (S1P) was also reported to promote inflammation in CLI[19]. Increasing attention is now being paid to the fact that the bioactive functions of CER can vary depending on the acyl chains[14]. It was recently discovered that BA-regulated intestinal FXR signaling suppresses CER production[20], suggesting a plausible NR-mediated signaling interplay between CER and BA metabolism. However, the specific roles of individual CERs in regulating CLI remain poorly understood.

CERs are generated through de novo synthesis, complex sphingolipid degradation, and salvage pathway[14]. Once generated, CERs are degraded by ceramidases, which are encoded by ASAH1, ASAH2, ACER1, ACER2, and ACER3[21]. Alkaline ceramidase 3 (ACER3) specifically hydrolyzes unsaturated CERs, particularly unsaturated-long-chain CERs (ULCCs), such as CER(d18:1/18:1)[22,23]. Our previous studies have implicated ACER3 dysregulation in liver diseases[24–27]. However, the function of ACER3 and ULCCs in regulating liver pathogenesis remains unclear.

In this work, we show that ACER3 expression is upregulated in cholestatic livers and positively correlates with the severity of CLI in patients. Targeting ACER3 increases CER(d18:1/18:1) in the cholestatic liver, which binds to LXRβ and activates its signaling transduction to enhance BA sulfation and restore lipogenesis. These mechanisms mitigate BA overload in hepatocytes and attenuate CLI. Our findings highlight the therapeutic potential of targeting ACER3 and CER(d18:1/18:1) to attenuate CLI.

## Results
### ACER3 plays a pathogenic role in CLI
To investigate the clinical relevance of CER dysregulation in CLI, we compared CERs and their metabolic enzymes in liver tissues from patients with non-CLI and CLI. Patients' characteristics are shown in Tables S1 and S2. Cholestasis significantly decreased hepatic CER(d18:1/26:0) and CER(d18:1/24:0), while increasing other saturated and unsaturated CERs, including CER(d18:1/18:1) and CER(d18:1/20:1) (Fig. 1a). Cholestasis also significantly decreased sphingosine (SPH) without affecting S1P (Fig. S1a). The mRNA levels of ACER3, B4GALT6, SGMS2, GLA, ASAH1, UGCG, and DEGS2 in the CER metabolic pathway

were upregulated in the liver tissues of patients with CLI (Fig. 1b, Fig. S1c). Correlation analysis demonstrated that ACER3 and B4GALT6 mRNA levels showed positive associations with various serum CLI severity markers (SCSMs) across all of the included patients (Fig. S1b). In patients with CLI, ACER3 mRNA levels exhibited the strongest positive correlations with a broader range of SCSMs compared to other enzymes, including direct bilirubin (DBIL), total bilirubin (TBIL), C-reactive protein (CRP), alkaline phosphatase (ALP), total bile acid (TBA), aspartate aminotransferase (AST), and alanine aminotransferase (ALT) (Fig. 1c). Interestingly, CER(d18:1/18:1), a specific substrate of ACER3[22,23], exhibited significant negative correlations with SCSMs in patients with CLI, including DBIL, TBIL, AST, and ALT (Fig. 1c). Besides, B4GALT6, SGMS2, GLA, and DEGS2 mRNA levels and CER(d18:1/18:0) exhibited weaker correlations with SCSMs in patients with CLI (Fig. 1c). Consistent with the mRNA levels, the protein levels of ACER3 and DEGS2 were significantly increased in CLI liver tissues compared to non-CLI liver tissues (Fig. 1d, e, Fig. S1d). The protein levels of ACER3 positively correlated with SCSMs in CLI patients (Fig. 1f). Online data mining suggested that cholestasis might upregulate ACER3 via transcription factors SP1, EGR1, and STAT3, which were determined as potential regulators of ACER3 expression (Fig. S1g) and known to be activated by cholestasis[28–30]. Given the significant upregulation of ACER3 in the cholestatic liver and its strong positive correlation with SCSMs, along with the inverse correlation of its substrate, CER(d18:1/18:1), with these markers, we investigated the potential pathogenic role of ACER3 in CLI.

To elucidate the role of ACER3 in CLI, we generated a mouse CLI model by BDL (Fig. S2a). Consistent with patient observations, Acer3 was upregulated in hepatocytes in mice with BDL-induced CLI (Fig. S2b−S2f). Additionally, BDL induced upregulation of mRNA levels for Smpd3, Sphk1, Cers3, Degs2, B4galt6, and Sgpp2 in the CER metabolic pathway (Fig. S2g, S2h), with significant increases in protein levels of Smpd3 and Cers3 (Fig. S2i, S2j). We established hepatocyte-specific Acer3 deletion mice (Fig. S2k−S2m) and investigated the effects of hepatocyte-specific (Acer3^ΔHep) and global deletion of Acer3 (Acer3^-/-) on CLI outcomes. Neither deletion negatively impacted the liver under basal conditions (Fig. S2n)[23]. Under BDL conditions, hepatic necrotic foci were significantly reduced in Acer3^ΔHep female mice compared to littermate controls (Fig. 1g, h). Correspondingly, the BDL-induced elevation of hepatocyte injury markers, transaminases, was remarkably mitigated in Acer3^ΔHep female mice (Fig. 1i). Acer3^ΔHep female mice also exhibited attenuated liver inflammation after BDL, evidenced by reduced expression of inflammatory genes (Fig. 1k) and decreased inflammatory infiltration (Fig. 1j, l, m). BDL-induced collagen deposition was attenuated in Acer3^ΔHep female mice (Fig. 1n), along with decreased fibrosis markers, including Col1a1, Col3a1, and αSMA (Fig. 1n−p). Similarly, Acer3^-/- female mice exhibited attenuated CLI (Fig. S3a−S3j). However, no substantial difference in CLI was observed between Acer3^-/- and Acer3^+/+ male mice (Fig. S3k−S3s), and male mice developed more severe CLI than female mice (Fig. S3). These findings demonstrate that Acer3 ablation attenuates CLI in female mice, indicating that ACER3 upregulation plays a pathogenic role in CLI.

### Acer3 ablation improves Sult2a1-mediated BA detoxification and normalizes BA metabolism to mitigate BA overload in hepatocytes with CLI
Hepatic transcriptome analysis followed by qPCR validation revealed that Sult2a families were markedly upregulated in Acer3^-/- female mice compared to Acer3^+/+ female mice (Fig. 2a, Fig. S4a). Sult2a1 is the most abundant member of the Sult2a family[31]. Sult2a1 mRNA levels were found to be decreased in the liver of control mice after BDL (Fig. 2b, Fig. S4b, S4i). However, Sult2a1 mRNA and protein levels were increased in hepatocytes of Acer3^ΔHep (Fig. 2b−d) and Acer3^-/- (Fig. S4b−S4e) female mice compared to littermate controls under basal and BDL conditions. Sult2a1 expression was lower in the liver of male mice

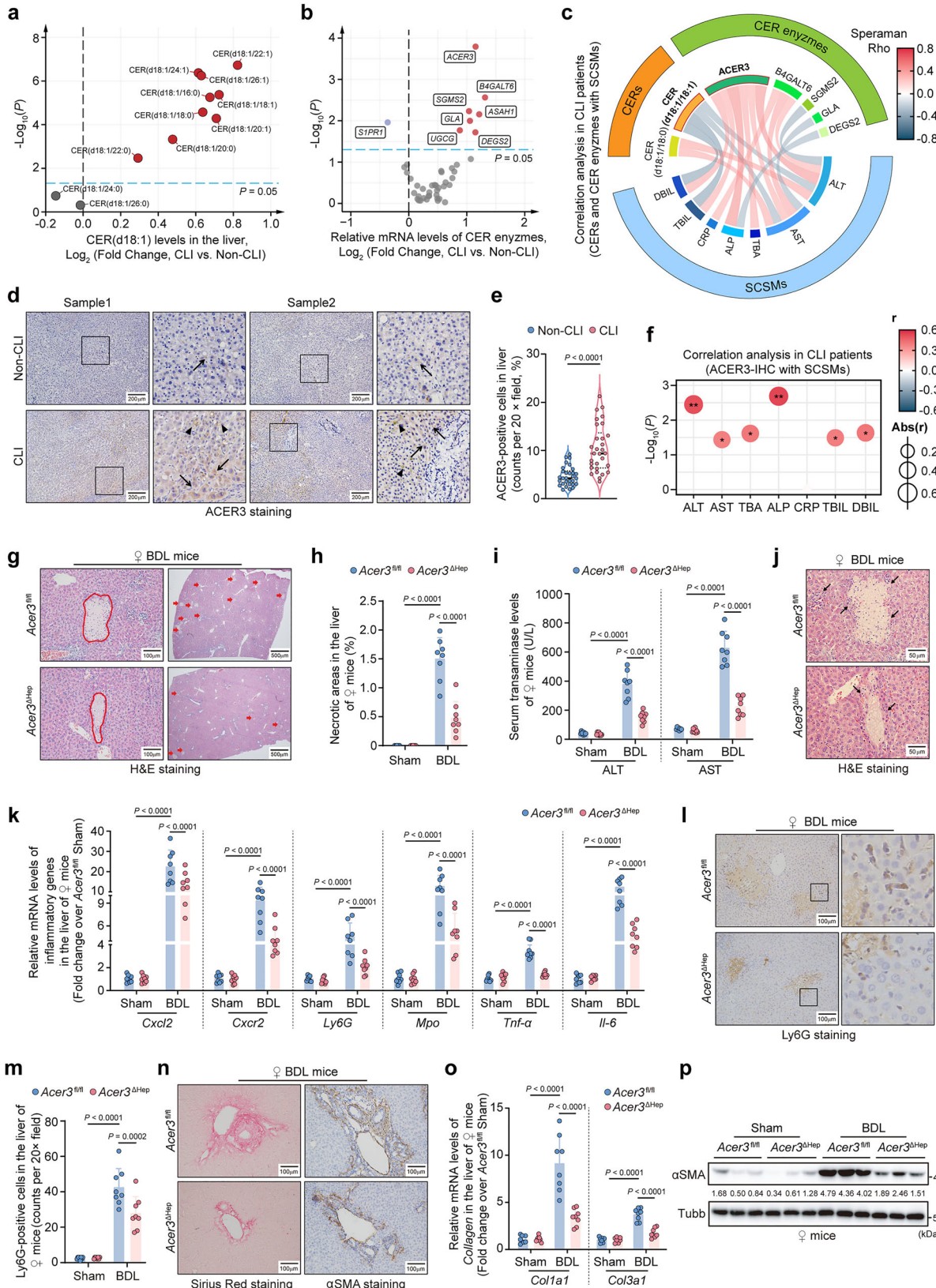

than in female mice (Fig. S4f), and *Acer3* deletion failed to alter hepatic *Sult2a1* expression in male mice (Fig. S4g–S4l).

SULTs catalyze BA sulfation in hepatocytes to form hydrophilic BA-sulfates, which are excreted by the kidney[10]. Since *Acer3* ablation upregulates SULTs, particularly *Sult2a*, we investigate if hepatocyte-specific *Acer3* ablation attenuates CLI through Sult2a1-catalyzed BA

sulfation in hepatocytes. We found that BA-sulfates were increased in the liver of *Acer3*^ΔHep female mice compared to *Acer3*^fl/fl female mice under basal and BDL conditions (Fig. 2e), including CA-sulfate, TLCA-sulfate, TCA-sulfate, GDCA-sulfate, TCDCA-sulfate, and total BA-sulfates (Fig. S5a). Consequently, BA-sulfates were increased in the serum and kidney in *Acer3*^ΔHep female mice after BDL (Fig. 2f, Fig. S5b,

**Fig. 1 | Hepatic *ACER3* is upregulated in CLI patients and hepatocyte-specific *Acer3* ablation attenuates CLI in female mice. a–f** Clinical relevance of ceramide (CER) dysregulation in cholestatic liver injury (CLI). Hepatic CERs in patients with CLI ($n = 30$) and non-CLI ($n = 30$) (**a**). mRNA levels of CER metabolic enzymes (**b**). Correlation of dysregulated CERs and their metabolic enzymes with serum CLI severity markers (SCSMs) in CLI patients (**c**). Hepatic alkaline ceramidase 3 (ACER3) staining (**d**) and quantification of ACER3-positive cells (**e**), the black arrows and black arrowheads indicate ACER3-positive cells and cholestasis, respectively. Correlation between ACER3-positive cells and SCSMs in CLI patients (**f**). **g–p** CLI in *Acer3*[fl/fl] and *Acer3*[ΔHep] female mice ($n = 8$). Hematoxylin and eosin (H&E) staining with circle areas and red arrows indicating necrotic foci (**g**) and quantification of necrotic areas (**h**). Serum transaminase levels (**i**). Higher-magnification images highlighting necrotic areas in H&E staining of Fig. 1g, with black arrows indicating regions of inflammatory infiltration (**j**). Inflammatory gene mRNA levels (**k**). Lymphocyte antigen 6 complex locus G6D (Ly6G) staining (**l**) and quantification of Ly6G-positive cells (**m**). Sirius Red (left panel) and alpha-smooth muscle actin (αSMA) staining (right panel) (**n**). Collagen mRNA levels (**o**). αSMA immunoblot (**p**). Data are expressed as mean ± SD. Statistical significances were tested by the unpaired two-sided Student's *t*-test (**a**, **b**, **e**), Spearman correlation test (**c**, **f**), and one-way ANOVA with Tukey's multiple comparisons (**h**, **i**, **k**, **m**, **o**). Source data are provided as a Source Data file.

S5c). Accumulation of BAs was reduced in the liver of *Acer3*[ΔHep] female mice after BDL (Fig. 2g, Fig. S5d). *Acer3* ablation also prevented BDL-induced decrease in mRNA levels of *Cyp7a1*, *Cyp27a1*, *Srd5b1*, *Baat*, and *Ntcp* in the liver of female mice (Fig. S5e, S5f), which are involved in BA synthesis and transportation[5]. In male mice, *Acer3* deletion failed to alter BA-sulfates or BAs and had minor effects on BA metabolizing enzymes in the liver (Fig. S5g–S5j). Next, *Sult2a1* expression was silenced to validate the function of *Sult2a1* in promoting BA sulfation in *Acer3*[fl/fl] and *Acer3*[ΔHep] female mice after BDL. Knockdown of *Sult2a1* in *Acer3*[fl/fl] and *Acer3*[ΔHep] female mice was validated at both mRNA and protein levels (Fig. 2h, Fig. S6a–S6c). *Sult2a1* knockdown in *Acer3*[fl/fl] female mice significantly reduced BA-sulfates (Fig. 2i, j, Fig. S6d–S6f) and augmented the accumulation of BAs (Fig. 2k, Fig. S6g), thereby exacerbating CLI, as evidenced by worsened hepatic necrosis, inflammation, and fibrosis (Fig. 2l–s, Fig. S6h, S6i). These results underscore the protective role of Sult2a1-catalyzed BA sulfation against CLI in female mice. Importantly, *Sult2a1* knockdown in *Acer3*[ΔHep] female mice abolished the enhanced BA sulfation and alleviated CLI, resulting in a similar degree of CLI as observed in *Sult2a1*-knockdown *Acer3*[fl/fl] female mice (Fig. 2h–s, Fig. S6d–S6i). These findings demonstrate that loss of *Sult2a1* abrogates the protective function of *Acer3* ablation against CLI in female mice, suggesting that *Acer3* ablation alleviates CLI through upregulating Sult2a1-catalyzed BA sulfation and normalizing BA metabolism in female mice.

### Lxrβ activation amplifies *Sult2a1* transcription in *Acer3*-deficient hepatocytes and attenuates CLI

Transcription of *Sult2a1* is controlled by NRs[32]. Given that *Acer3* ablation amplifies *Sult2a1* transcription, we investigated whether NRs mediate this amplification. We found that *Lxrβ* expression was significantly increased in the liver of *Acer3*[ΔHep] female mice compared to *Acer3*[fl/fl] female mice under basal and BDL conditions (Fig. 3a–c). *Acer3* ablation failed to impact *Car*, *Erα*, *Erγ*, *Fxr*, *Lxrα*, *Ppary*, *Pxr*, or *Vdr* (Fig. S6j, S6k). Besides *Lxrβ*, *Acer3* ablation also increased Rxrα protein levels in female mice after BDL (Fig. S6k). The cytoplasmic and nuclear Lxrβ were increased in the liver of *Acer3*[ΔHep] female mice under basal conditions, with sustained nuclear Lxrβ levels after BDL (Fig. 3d–f). Importantly, *Lxrβ* knockdown decreased nuclear Lxrβ and suppressed *Sult2a1* upregulation in the hepatocytes of *Acer3*[ΔHep] female mice after BDL (Fig. 3g–k). Consequently, *Lxrβ* knockdown decreased BA-sulfates in the liver, serum, and kidney (Fig. 3l, m, Fig. S6l–S6n), exaggerating CLI in *Acer3*[ΔHep] female mice (Fig. 3n–v). These results revealed that *Lxrβ* is essential for *Acer3* ablation to amplify *Sult2a1* transcription and attenuate CLI in female mice.

### *Acer3* ablation increases CER(d18:1/18:1) and restores Lxrβ-driven lipogenesis in the cholestatic liver

To investigate the impact of hepatocyte-specific *Acer3* ablation on CER metabolism in the cholestatic liver, we found BDL decreased CER(d18:1/22:1) while increasing CER(d18:1/18:1), CER(d18:1/20:1), CER(d18:1/24:1), CER(d18:1/26:1), and total unsaturated CERs in the liver of female mice (Fig. 4a, Fig. S7a). CER(d18:1/18:1) and CER(d18:1/20:1) were increased in the liver of *Acer3*[ΔHep] female mice under basal

conditions, with CER(d18:1/18:1) further increased after BDL (Fig. 4a). Hepatic SPH was decreased by BDL, SPH or S1P were not affected by *Acer3* ablation (Fig. S7b). Since *Acer3* deficiency attenuated BA overload and upregulated *Lxrβ* expression, which is implicated in regulating lipid metabolism[33], we investigated if *Acer3* ablation affects lipidome in the cholestatic liver of female mice. Measurements of the mRNA levels of Lxrβ-associated genes involved in lipid metabolism demonstrated that BDL reduced stearoyl-coenzyme A desaturase 1 (Scd1) and carbohydrate-responsive element-binding protein (Chrebp) mRNA levels in the liver of *Acer3*[fl/fl] female mice (Fig. 4b). In the liver of *Acer3*[ΔHep] female mice, the mRNA levels of *Scd1* and fatty acid synthase (Fasn) were increased under basal conditions and maintained at higher levels after BDL compared to those in *Acer3*[fl/fl] female mice (Fig. 4b). Lipidomics revealed that BDL substantially decreased lipid content in the liver of *Acer3*[fl/fl] female mice, particularly affecting triglycerols (TGs) and phospholipids (PLs) (Fig. 4c–e), while this reduction was partially reversed in *Acer3*[ΔHep] female mice, with smaller decreases in TGs and PLs (Fig. 4c–h). ORO staining showed that the BDL-induced reduction of hepatic lipids was reversed in *Acer3*[ΔHep] female mice (Fig. 4i, j), and this effect was abolished by either *Sult2a1* or *Lxrβ* knockdown (Fig. 4k, l). *Lxrβ* knockdown also downregulated *Scd1* and *Fasn* in *Acer3*[ΔHep] female mice after BDL (Fig. S7c). These data suggest that *Acer3* ablation specifically increases CER(d18:1/18:1) and restores Lxrβ-driven lipogenesis in the cholestatic liver of female mice.

### CER(d18:1/18:1) upregulates *Sult2a1* and improves lipogenesis to attenuate CLI by activating Lxrβ signaling transduction

Since *Acer3* ablation specifically increased CER(d18:1/18:1) in the cholestatic liver, we determined whether CER(d18:1/18:1) mediates the protective function of *Acer3* ablation against CLI. To this end, we evaluated the effects of CER(d18:1/18:1) treatment on CLI in C57BL/6 J wild-type (WT) female mice. We found that CER(d18:1/18:1) treatment led to a prominent elevation in hepatic CER(d18:1/18:1) levels, with only minor increases in CER(d18:1/18:0) and CER(d18:1/20:1) (Fig. 5a, Fig. S8a). CER(d18:1/18:1) treatment significantly attenuated CLI in female mice (Fig. 5b–j). CER(d18:1/18:1) treatment also increased nuclear Lxrβ (Fig. 5k–n) and *Sult2a1* expression in hepatocytes after BDL (Fig. 5n–p). Consequently, CER(d18:1/18:1) treatment increased BA-sulfates in the liver, serum, and kidney after BDL (Fig. 5q, r, Fig. S8b–S8d). ORO staining demonstrated that CER(d18:1/18:1) treatment increased hepatic lipids after BDL (Fig. 5s, t). Accordingly, the mRNA levels of *Scd1* and *Fasn* were increased under CER(d18:1/18:1) treatment on CLI (Fig. 5u), while CER(d18:1/18:1) treatment did not significantly impact the expressions of *Srebp1* and *Pparα* in the liver after BDL (Fig. S8e, S8f). These findings uncover a protective role of CER(d18:1/18:1) in CLI.

LXRβ is activated by binding lipid ligands[8]. Given that *Acer3* ablation upregulated *Lxrβ*, we examined its influence on endogenous ligands of LXRβ[34]. However, none of the detected Lxrβ ligands differed significantly between *Acer3*[ΔHep] and *Acer3*[fl/fl] female mice under basal or BDL conditions (Fig. S9). We then determined if CER(d18:1/18:1) functions as a ligand of Lxrβ. In silico docking analysis demonstrated that CER(d18:1/18:1) bound Lxrβ ligand-binding domain (LBD) primarily

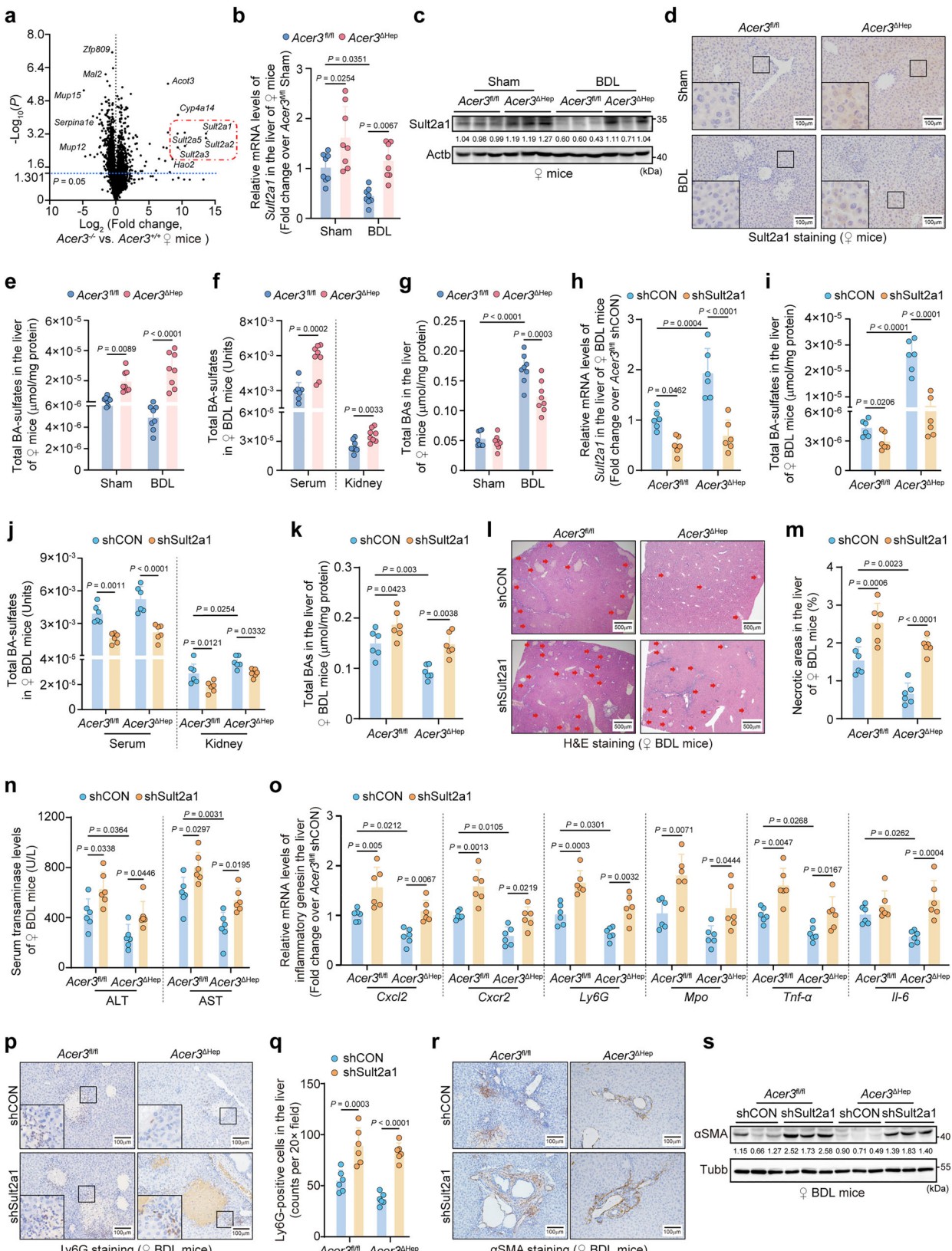

through hydrophobic interaction, displaying a moderate predicted affinity (Fig. 6a, Fig. S8g–S8i). Upon activation by agonists, LXRβ translocates to the nucleus in cells[35]. Measurement of CERs in nuclear components demonstrated increased nuclear CER(d18:1/18:1) levels in the liver of *Acer3*[ΔHep] female mice compared to Acer3[fl/fl] female mice under basal conditions, and further augmentation after BDL (Fig. 6b).

This confirmed the nuclear colocalization of CER(d18:1/18:1) and Lxrβ, supporting the interaction of CER(d18:1/18:1)-Lxrβ. Functional experiments demonstrated that *Lxrβ* knockdown inhibited the CER(d18:1/18:1)-induced upregulation of Sult2a1-catalyzed BA sulfation (Fig. 6c–g, Fig. S8j) and impeded the restoration of lipogenesis possibly via *Scd1* and *Fasn* (Fig. 6h–j), eventually diminished the protective effects of

**Fig. 2 | *Acer3* ablation improves Sult2a1-catalyzed BA sulfation to attenuate CLI. a** Differentially expressed genes (DEGs) of hepatic transcriptomes in *Acer3*^(+/+) and *Acer3*^(-/-) female mice (*n* = 4). **b–g** Sulfotransferase 2A1 (Sult2a1)-catalyzed bile acid (BA) sulfation in *Acer3*^(fl/fl) and *Acer3*^(ΔHep) female mice (*n* = 8). mRNA (**b**) and protein levels of *Sult2a1* (**c** and **d**). Total BA-sulfates in the liver (**e**), serum (μmol/ml) (**f**), and kidney (μmol/mg) (**f**). Total BAs in the liver (**g**). **h–s** Sult2a1-catalyzed BA sulfation and CLI in the liver of *Acer3*^(fl/fl) and *Acer3*^(ΔHep) BDL female mice with or without *Sult2a1* knockdown (*n* = 6). *Sult2a1* mRNA levels (**h**). Total BA-

sulfates in the liver (**i**), serum (μmol/ml) (**j**), and kidney (μmol/mg) (**j**). Total BAs in the liver (**k**). H&E staining with the red arrows indicating necrotic foci (**l**) and quantification of necrotic areas (**m**). Serum transaminase levels (**n**). Inflammatory gene mRNA levels (**o**). Ly6G staining (**p**) and quantification of Ly6G-positive cells (**q**). αSMA staining (**r**). αSMA immunoblot (**s**). Data are expressed as mean ± SD. Statistical significances were tested by the unpaired two-sided Student's *t*-test (**a**, **f**) and one-way ANOVA with Tukey's multiple comparisons (**b**, **e**, **g-k**, **m-o**, **q**). Source data are provided as a Source Data file.

CER(d18:1/18:1) against CLI in WT female mice (Fig. 6k−s). LXRα, an isotype of LXRβ, is also known as a transcriptional regulator of SULT2A1[36]. To clarify the role of Lxrβ in mediating the protective effects of *Acer3* ablation against CLI via CER(d18:1/18:1), we examined the impact of *Acer3* ablation and *Lxrβ* knockdown on *Lxrα* expression in vivo. We found that combined *Acer3* ablation and *Lxrβ* knockdown had no significant effect on *Lxrα* expression in mouse livers, suggesting that Lxrα is unlikely involved in the Acer3-mediated regulation of *Sult2a1* (Fig. S8k). These results demonstrated that CER(d18:1/18:1) activates Lxrβ to upregulate Sult2a1-catalyzed BA sulfation and attenuate CLI in female mice, uncovering CER(d18:1/18:1) as a mediator of *Acer3* ablation in attenuating CLI.

### ACER3 knockdown promotes CER(d18:1/18:1)-LXRβ signaling transduction to attenuate CLI in human liver-derived cells

To translate our animal findings to a human context, we first examined sex-specific differences in *SULT2A1* expression and CER-metabolizing enzymes in human liver tissues. In silico analysis using publicly available datasets from the Genotype-Tissue Expression (GTEx) Portal (https://www.gtexportal.org/) revealed that the mRNA expression of hepatic *SULT2A1* did not significantly differ between female and male healthy humans (Fig. 7a). We further evaluated sex-specific differences in *SULT2A1* expression using collected human liver tissues. Patients' characteristics disaggregated for sex are shown in Table S2. Consistently, measurements of mRNA (Fig. 7b) and protein (Fig. 7c−e) levels of *SULT2A1* in the collected liver tissues demonstrated no significant sex-specific differences in *SULT2A1* expression in patients with or without CLI. In the liver tissues with CLI, *SULT2A1* expression was significantly reduced compared to the non-CLI liver tissues, with no differences between sexes (Fig. 7b, f, g), indicating that cholestasis downregulated hepatic *SULT2A1* similarly across both sexes. Regarding CER-metabolizing enzymes, subgroup analyses comparing CLI and non-CLI groups revealed that cholestasis significantly upregulated the mRNA levels of *ACER3*, *ASAH1*, *GLA*, *B4GALT6*, and *SGMS2* in male patients, while in female patients, it significantly increased the mRNA levels of *ACER3* and *DEGS2* (Fig. 7h, Fig. S10a). Further analyses comparing males and females within the same condition using both in silico GTEx Portal datasets and the collected liver tissues, revealed no sex-specific differences in the mRNA levels of CER-related enzymes in either healthy or cholestatic liver tissues (Fig. 7i, j). These findings confirm that the upregulation of hepatic *ACER3* by cholestasis is not sex-specific in humans. Comparisons of hepatic CER levels between CLI and non-CLI groups showed that cholestasis significantly increased most CER species in male patients, except for CER(d18:1/24:0) and CER(d18:1/26:0) (Fig. S10b). Similarly, in female patients, cholestasis significantly increased most CER species, excluding CER(d18:1/22:0) and CER(d18:1/24:0) (Fig. S10c). Comparisons of hepatic CER levels between sexes within the same condition indicated that most CER(d18:1) species did not differ significantly between males and females in non-cholestatic livers (Fig. 7k). Although CER(d18:1/26:0) and CER(d18:1/26:1) were lower in the female liver compared to male liver tissues with CLI (Fig. 7l), cholestasis-induced upregulation of CER(d18:1/18:1) was comparable between sexes (Fig. 7m). Similarly, cholestasis decreased SPH levels in both male and female patients without affecting S1P levels (Fig. S10d), and SPH and S1P levels showed

no significant differences between sexes in either non-CLI or CLI livers (Fig. S10e). These findings collectively suggest that while subtle sex-specific trends exist in the dysregulation of certain CER-metabolizing enzymes and specific CER species in response to cholestasis, the dysregulation of *SULT2A1* and *ACER3*, increases in CER levels, and changes in SPH/S1P levels are largely consistent between males and females.

Next, we established an in vitro CLI model by treating human liver-derived cell lines with toxic BA[37]. HepG2 cells, well-differentiated hepatocarcinoma cells derived from a male patient[38] with high *SULT2A1* expression[39], were selected for in vitro studies after validating their high *SULT2A1* expression levels (Fig. S11a). LCA treatment was applied to mimic BA overload during cholestasis (Fig. S11b). Consistent with in vivo findings, *ACER3* knockdown alleviated cell death with increasing SULT2A1, LCA-sulfate, and lipid content in HepG2 cells after LCA treatment (Fig. 8a−c, Fig. S11c−S11f). Silencing *SULT2A1* abolished these protective effects of *ACER3* knockdown (Fig. 8d, Fig. S11g, S11h). Furthermore, *ACER3* knockdown upregulated nuclear LXRβ in LCA-treated HepG2 cells (Fig. 8e, S11i). Luciferase reporter assays confirmed that *ACER3* knockdown enhanced *SULT2A1* promotor activity (Fig. 8f), and this effect was abolished after silencing *LXRβ* (Fig. 8g). Consequently, silencing *LXRβ* decreased *SULT2A1* expression levels (Fig, 8h, Fig. S11k) and diminished the protective effects of *ACER3* knockdown against LCA in HepG2 cells (Fig. 8h). These findings indicate that *ACER3* knockdown mitigates LCA-induced cell death through activation of LXRβ-driven SULT2A1-catalyzed sulfation.

CER measurement showed that LCA treatment increased saturated and unsaturated CERs in HepG2 cells (Fig. 8i, Fig. S12a). Notably, the increase in CER(d18:1/18:1) was specifically augmented by *ACER3* knockdown (Fig. 8i). In line with in vivo findings, CER(d18:1/18:1) treatment attenuated LCA-induced cell death while upregulating LXRβ and SULT2A1 (Fig. S12b−S12e), silencing *LXRβ* abolished the protective effects of CER(d18:1/18:1) against LCA in HepG2 cells (Fig. S12f−S12h). To verify CER(d18:1/18:1)-LXRβ interaction in a human setting, we found that *ACER3* knockdown significantly increased nuclear CER(d18:1/18:1) levels in LCA-treated HepG2 cells (Fig. S13a), confirming the nuclear colocalization of CER(d18:1/18:1) and LXRβ. Surface plasmon resonance (SPR) revealed that CER(d18:1/18:1) bond recombinant LXRβ in a dose-dependent manner (Fig. 8j), and immunoprecipitation confirmed CER(d18:1/18:1) presence in LXRβ protein (Fig. 8k, l, Fig. S13b), providing evidence for CER(d18:1/18:1)-LXRβ interaction. Virtual docking analysis revealed that CER(d18:1/18:1) bonded LXRβ LBD with the highest predicted affinity reaching a grid score of −117.30 (Fig. S13c and Table S3). Notably, CER(d18:1/18:1) was predicted to interact with key amino acid residues in LXRβ LBD (Fig. 8m, Fig. S13c, S13d) known for strong interactions with lipid agonists[40]. Particularly, CER(d18:1/18:1) interacted with His435 and Trp457 of LXRβ (Fig. 8n, Fig. S13e), which essentially mediate the ligand-mediated LXR agonism[41,42]. Nevertheless, we tested whether CER(d18:1/18:1) also interacted with LXRα, an isotype of LXRβ, and whether LXRα contributed to ACER3-mediated regulation of *SULT2A1* expression. Computational modeling revealed that CER(d18:1/18:1) exhibited a lower binding affinity for LXRα than LXRβ across all available protein structures, indicating a preference for LXRβ (Table S3, S4). Consistent with

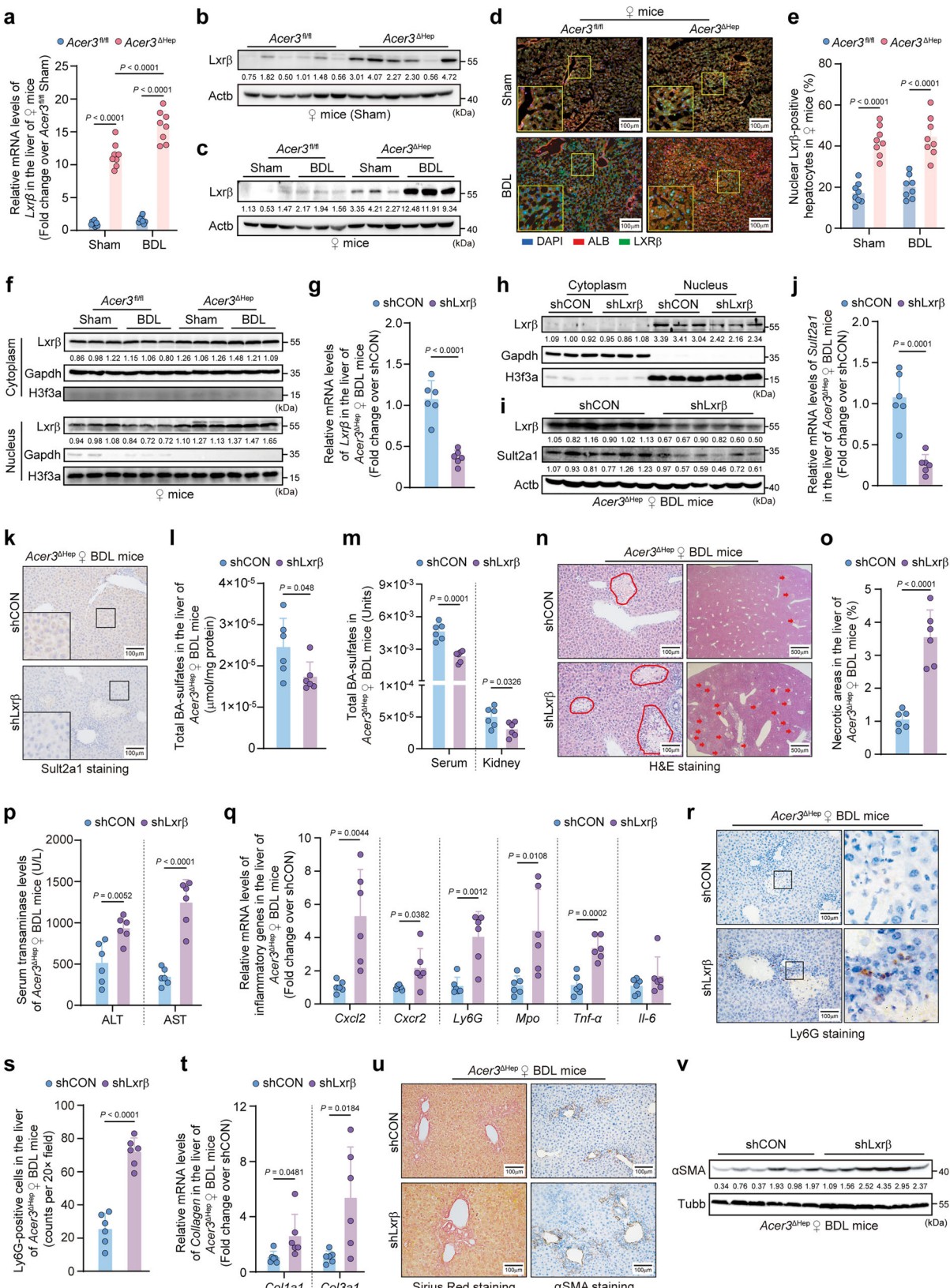

our in vivo data, double knockdown of *ACER3* and *LXRβ* had no significant effect on LXRα expression in HepG2 cells (Fig. S13f). Additionally, in *ACER3*-knockdown HepG2 cells, although *LXRα* knockdown led to a modest reduction in the mRNA levels of *SULT2A1*, this reduction was minimal and did not result in a significant decrease in protein expression of *SULT2A1*, in contrast to the more pronounced effects

observed with *LXRβ* knockdown (Fig. S13g–S13k), underscoring the pivotal role of LXRβ in mediating the effects of *ACER3* knockdown relative to LXRα. These findings unraveled CER(d18:1/18:1) as an endogenous agonist of LXRβ, indicating that CER(d18:1/18:1)-LXRβ signaling transduction crucially mediates the protective effects of *ACER3* ablation against CLI.

**Fig. 3 | *Acer3* ablation upregulates *Lxr*β to activate Sult2a1-catalyzed BA sulfation to attenuate CLI. a–f** Hepatic liver X receptor β (*Lxrβ*) expression in *Acer3*^fl/fl^ and *Acer3*^ΔHep^ female mice (*n* = 8). *Lxrβ* mRNA levels (**a**). Immunoblot of Lxrβ in female mice with sham operation (**b**). Immunoblot of Lxrβ in the liver of female mice under sham and bile duct ligation (BDL) conditions (**c**). Immunofluorescent co-staining with Lxrβ and albumin (Alb) (**d**) and quantification of nuclear-Lxrβ-positive hepatocytes (**e**). Immunoblot of cytoplasmic and nuclear Lxrβ (**f**). **g–v** Examination of Lxrβ-Sult2a1 pathway and CLI in the liver of *Acer3*^ΔHep^ female BDL mice with or without *Lxrβ* knockdown (*n* = 6). *Lxrβ* mRNA levels (**g**). Immunoblot of cytoplasmic and nuclear Lxrβ (**h**). Protein levels of Lxrβ and Sult2a1 (**i**). *Sult2a1* mRNA levels (**j**). Sult2a1 staining in

the liver (**k**). Total BA-sulfates in the liver (**l**), serum (μmol/ml) (**m**), and kidney (μmol/mg) (**m**). H&E staining with the circle areas and red arrows indicating necrotic foci (**n**) and quantification of necrotic areas (**o**). Serum transaminase levels (**p**). Inflammatory gene mRNA levels (**q**). Ly6G staining (**r**) and quantification of Ly6G-positive cells (**s**). Collagen mRNA levels (**t**). Sirius Red staining (left panel) and αSMA staining (right panel) (**u**). αSMA immunoblot (**v**). Data are expressed as mean ± SD. Statistical significances were tested by the unpaired two-sided Student's *t*-test (**g, j, l, m, o-q, s, t**) and one-way ANOVA with Tukey's multiple comparisons (**a, e**). Source data are provided as a Source Data file.

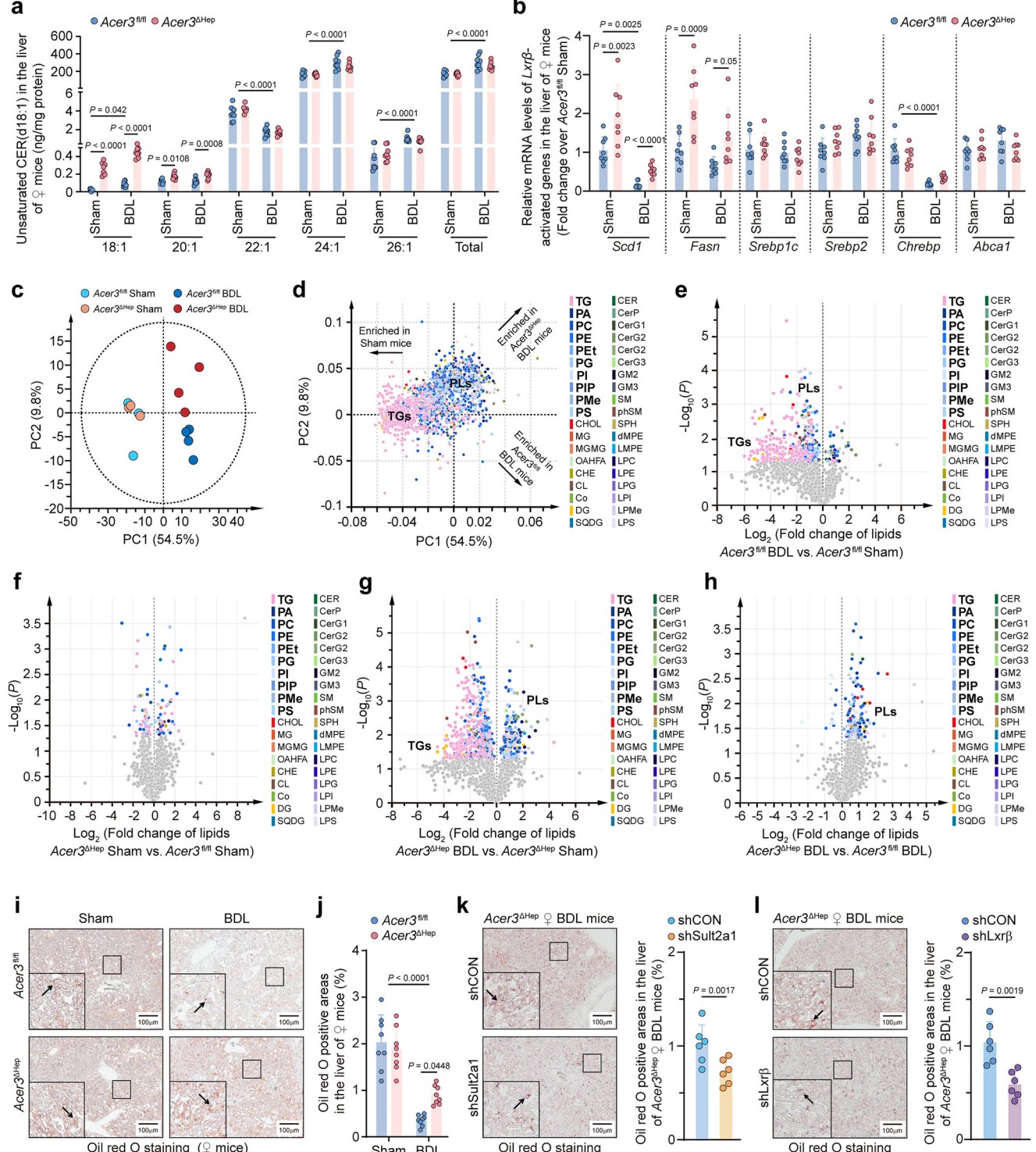

**Fig. 4 | *Acer3* ablation increases CER(d18:1/18:1) and improves lipogenesis in the cholestatic liver. a–j** Hepatic CERs and lipid content in *Acer3*^fl/fl and *Acer3*^ΔHep female mice (*n* = 8). Hepatic unsaturated CER(d18:1) (**a**). mRNA levels of Lxrβ-driven lipogenic genes in the liver (**b**). Principal component analysis (PCA) analysis of discrimination in hepatic lipid content with scoring plot (**c**) and loading plot (**d**) in *Acer3*^fl/fl and *Acer3*^ΔHep female mice under both sham and BDL conditions (*n* = 4), the arrow labels indicate the enriched lipid content in the indicated mouse groups. 35 classes of lipids were detected, including triglyceride (TG), phosphatidic acid (PA), phosphatidyl choline (PC), phosphatidylethanolamine (PE), phosphatidylethanol (PEt), phosphatidylglycerol (PG), phosphatidylinositol (PI), phosphatidylinositol phosphate (PIP), phosphatidylmethanol (PMe), phosphatidylserine (PS), choles-terol (CHOL), monoglyceride (MG), monogalactosylmonoacylglycerol (MGMG), (O-acyl)−1-hydroxy fatty acid (OAHFA), cholesteryl ester (CHE), cardiolipin (CL), coenzyme (Co), diglyceride (DG), sulfoquinovosyldiacylglycerol (SQDG), ceramide (CER), ceramides phosphate (CerP), monogylcosylceramide (CerG1), diglyco-sylceramide (CerG2), triglycosylceramide (CerG3), ganglioside (GM), sphingo-myelin (SM), phytosphingosine (phSM), sphingoshine (SPH), dimethylphosphatidylethanolamine (dMPE),

lysodimethylphosphatidylethanolamine (LMPE), lyso-phosphatidylcholine (LPC), lyso-phosphatidylethanolamine (LPE), lyso-phosphatidylglycerol (LPG), lyso-phosphatidylinositol (LPI), lyso-phosphatidylmethanol (LPMe), lyso-phosphatidylserine (LPS). Among them, PA, PC, PE, PEt, PG, PI, PIP, PMe, and PS were classified into PLs, TG and PLs were dramatically affected by BDL. Volcano plot exhibiting the difference of individual lipid species between *Acer3*^fl/fl female mice with BDL and sham operation (**e**), *Acer3*^ΔHep and *Acer3*^fl/fl female mice with sham operation (**f**), *Acer3*^ΔHep female mice with BDL and sham operation (**g**), *Acer3*^ΔHep and *Acer3*^fl/fl female mice with BDL operation (**h**). Oil Red O (ORO) staining with black arrows illustrating ORO-positive areas (**i**) and quantification of ORO-positive areas (**j**). (**k**) ORO staining with black arrows illustrating ORO-positive areas and quanti-fication of ORO-positive areas in the liver of *Acer3*^ΔHep BDL female mice with or without *Sult2a1* knockdown (*n* = 6). **l** ORO staining with black arrows illustrating ORO-positive areas and quantification of ORO-positive areas in the liver of *Acer3*^ΔHep BDL female mice with or without *Lxrβ* knockdown (*n* = 6). Data are expressed as mean ± SD. Statistical significances were tested by the one-way ANOVA with Tukey's multiple comparisons (**a**, **b**, **j**) and unpaired two-sided Student's *t*-test (**e**-**h**, **k**, **l**). Source data are provided as a Source Data file.

## Discussion

This study presents the insights regarding the role of ACER3 and its endogenous substrate, CER(d18:1/18:1), in regulating the metabolic resilience of BAs and lipids in the cholestatic liver. We found that cholestasis upregulates *ACER3*, while *ACER3* ablation promotes the binding of CER(d18:1/18:1) to LXRβ to activate LXRβ signaling, thereby improving BA detoxification and lipogenesis to attenuate CLI. These findings unravel the function of CER(d18:1/18:1)-LXRβ signaling in maintaining BA and lipid metabolic resilience to counter BA overload in hepatocytes, serving as a promising therapeutic target of CLI.

Dysregulation of CER metabolism is associated with cholestatic diseases[15–17,19]. In this study, we observed most CER species were increased in the cholestatic liver of patients (Fig. 1a). Additionally, enzymes involved in CER generation and degradation were both upregulated, indicating an overall activation of CER metabolism in response to cholestasis (Fig. 1b). Hepatic *ACER3* was found to be upregulated by cholestasis and positively correlated with CLI severity in patients with cholestasis (Fig. 1c). This upregulation of *ACER3* and its strong positive correlation with SCSMs underscored the potential pathogenic role of *ACER3* in CLI. Accordingly, our study demonstrated that *Acer3/ACER3* ablation attenuated CLI in female mice and HepG2 cells (Figs. 1 and 8), indicating that *ACER3* upregulation promotes CLI. Our online data mining suggested that transcription factors SP1, EGR1, and STAT3, which are known to be activated by cholestasis[28–30] and regulate various enzyme expressions in CER metabolism[43–47], might act as the transcriptional regulators of *ACER3* expression (Fig. S1g). These suggest that cholestasis may drive *ACER3* upregulation through mul-tiple regulatory pathways involving these transcription factors. These findings provide a plausible explanation for the upregulation of *ACER3* in cholestasis and reinforce the role of *ACER3* in CER metabolism under cholestatic conditions.

SULTs-catalyzed BA sulfation is essential for BA detoxification to reduce their accumulation and toxicity[10]. Our findings demonstrated that hepatocyte-specific *Acer3* deletion upregulated Sult2a genes and increased BA-sulfates in the liver, serum, and kidney of female mice with CLI (Fig. 2b–f). Moreover, hepatocyte-specific *Acer3* deletion prevented the dysregulation of BA metabolizing enzymes induced by BDL (Fig. S5). These actions collectively improve the detoxification and elimination of BAs, thereby reducing BA toxicity and accumulation in the cholestatic liver. Upon silencing *Sult2a1*, the most abundant member of the *Sult2a* family, we observed a decrease in BA-sulfates and exaggerated CLI in the control and *Acer3* deficient female mice (Fig. 2h–s). Similarly, *ACER3* knockdown upregulated SULT2A1-catalyzed BA sulfation to alleviate LCA toxicity, while *SULT2A1* silen-cing abolished these effects in HepG2 cells (Fig. 8). These findings confirm that *ACER3* ablation improves BA detoxification by

upregulating *SULT2A1* to attenuate CLI, uncovering a function of CER metabolism in regulating BA detoxification.

NRs are key transcriptional factors of the *Sult2a1/SULT2A1* gene and critically regulate lipid homeostasis[32,33]. We found that hepatocyte-specific *Acer3* deletion upregulated *Lxrβ* expression, while *Lxrβ* knockdown suppressed Sult2a1-catalyzed BA sulfation and exag-gerated CLI in *Acer3*-deficient female mice (Fig. 3). Furthermore, *ACER3* knockdown increased *SULT2A1* promoter activity in HepG2 cells, which was abolished by *LXRβ* knockdown, confirming that *ACER3* regulates SULT2A1 through LXRβ. Besides *Lxrβ*, *Rxrα* was also upre-gulated by *Acer3* ablation (Fig. S6k). Since Lxrβ forms obligate het-erodimers with Rxrα for downstream transcriptional function[35], Rxrα may also contribute to *Sult2a1* upregulation after *Acer3* ablation. Consistent with our findings, Hirdesh Uppal et al. reported that Lxrα activation attenuates CLI by enhancing BA sulfation in female mice, while *Lxrα* and *Lxrβ* double deletion increases cholestatic sensitivity[36]. Adding to these findings, our study reinforces the protective function of *Lxrβ* activation against CLI by enhancing BA detoxification. Notably, LXRα, an isotype of LXRβ, is also recognized as a transcriptional reg-ulator of SULT2A1[36]. We found that *Lxrα* expression was not affected by combined *Acer3* ablation and *Lxrβ* knockdown in female mouse livers (Fig. S8k). Additionally, *LXRα* knockdown was insufficient to sig-nificantly reduce the protein expression of *SULT2A1* in *ACER3*-knock-down HepG2 cells (Fig. S13g−S13k), indicating that LXRα does not play a primary role in mediating the upregulation of *SULT2A1* expression induced by *ACER3* knockdown. These findings indicate that the reg-ulation of *SULT2A1* expression by ACER3 is primarily mediated through LXRβ.

ACER3 is known for its specific hydrolysis of ULCCs, particularly CER(d18:1/18:1)[22,23]. Despite the upregulation of *ACER3*, its substrate CER(d18:1/18:1) was also found to be increased in the cholestatic liver of patients and mice (Figs. 1a, 4a). This paradoxical increase may result from the initial accumulation of CER(d18:1/18:1) from CER generation through CER synthesis and complex sphingolipid degradation in the cholestatic liver (Figs. S1c, S2h). We found that enzymes contributing to CER production were upregulated by cholestasis, including DEGS2 in humans and Smpd3 and Cers3 in mice (Figs. S1, S2), indicating activation of CER generation by cholestasis. Concurrently, the upre-gulation of *ACER3/Acer3* and *ASAH1* suggested activation of CER degradation. However, while overall CER levels were increased in response to cholestasis, the levels of SPH, a CER degradation product, were significantly decreased (Figs. S1a, S7b). This suggests that while CER generation was upregulated, the activation of degradation cata-lyzed by ACER3/Acer3 and ASAH1 might have been insufficient to significantly affect SPH levels, or SPH may have been utilized as a substrate for the salvage synthesis of CERs[14], leading to decreased SPH

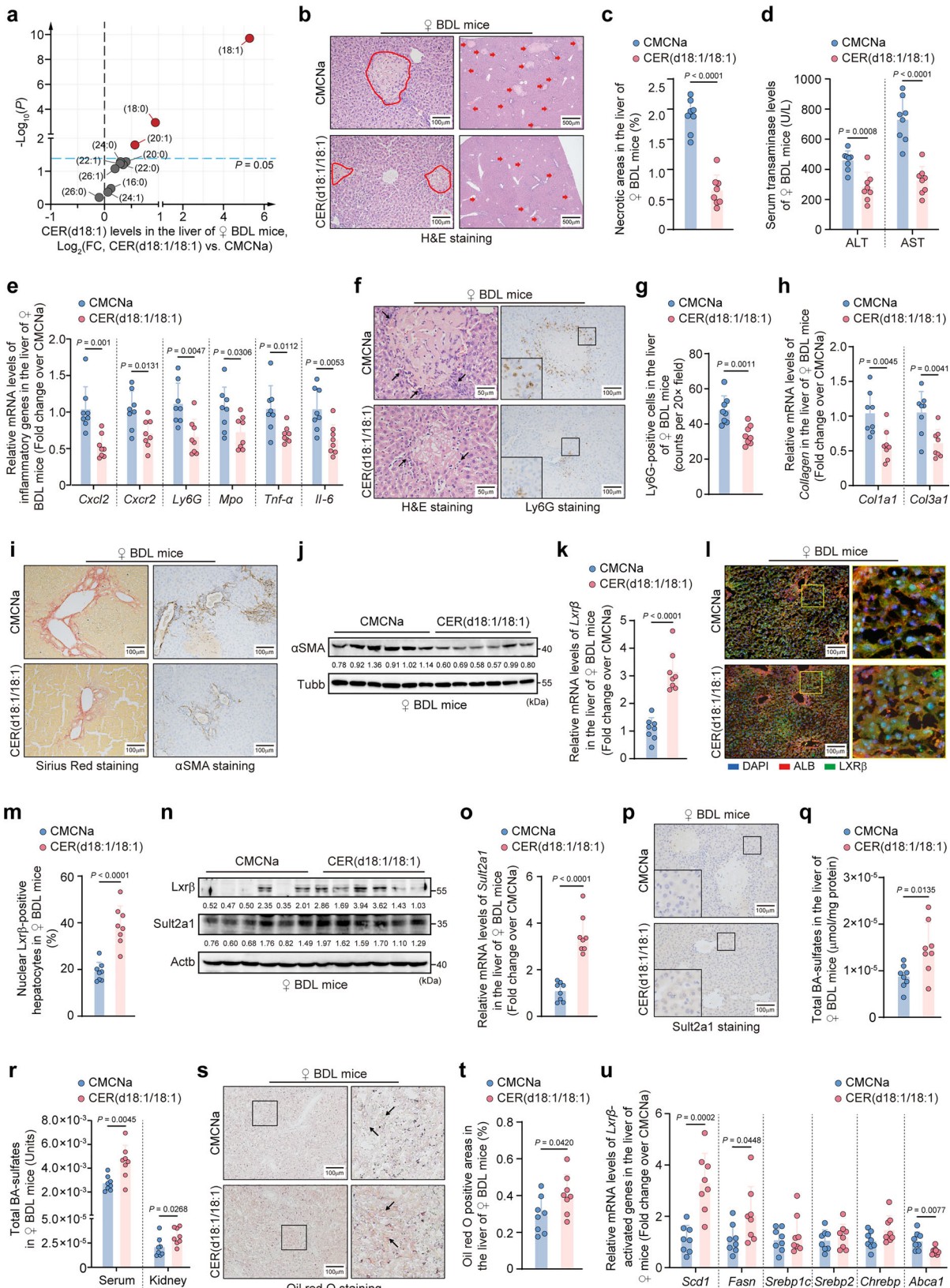

levels. Therefore, CER generation likely predominated over degradation in the cholestatic liver. Consequently, although *Acer3* was upregulated to hydrolyze CER(d18:1/18:1), the overwhelming production of CER(d18:1/18:1) exceeded its hydrolytic capacity, resulting in net accumulation of CER(d18:1/18:1) in the cholestatic liver. Our observation that *Acer3* ablation further increased CER(d18:1/18:1) in the

cholestatic liver (Fig. 4a) underscores this dynamic, indicating that *ACER3* upregulation impedes the buildup of CER(d18:1/18:1) in the cholestatic liver.

In addition to CER dysregulation, cholestasis also disrupts lipogenesis, resulting in the reduction of hepatic lipids[48,49]. Surprisingly, *Acer3* ablation substantially reserved this lipid reduction (Fig. 4). This

**Fig. 5 | CER(d18:1/18:1) upregulates Lxrβ and Sult2a1-catalyzed BA sulfation to attenuate CLI.** CER(d18:1/18:1)-treated C57BL/6 J wild-type (WT) female mice were subjected to BDL (n = 8). **a** Hepatic CER levels after CER(d18:1/18:1) treatment. **b–j** Examination of CLI. H&E staining with the circle areas and red arrows indicating necrotic foci (**b**) and quantification of necrotic areas (**c**). Serum transaminase levels (**d**) Inflammatory gene mRNA levels (**e**). Higher-magnification images highlighting necrotic areas in H&E staining of Fig. 5b, with black arrows indicating regions of inflammatory infiltration. (**f**, left panel). Ly6G staining (**f**, right panel), and quantification of Ly6G-positive cells (**g**). Collagen mRNA levels (**h**). Sirius Red staining (left panel) and αSMA staining (right panel) (**i**). αSMA immunoblot (**j**). **k–s** Examination of Lxrβ-Sult2a1 pathway. *Lxrβ* mRNA levels in the liver (**k**). Immunofluorescent co-staining with Lxrβ and Alb (**l**) and quantification of nuclear-Lxrβ-positive hepatocytes (**m**). Immunoblot of Lxrβ and Sult2a1 (**n**). *Sult2a1* mRNA levels (**o**). Sult2a1 staining in the liver (**p**). Total BA-sulfates in the liver (**q**), serum (μmol/ml) (**r**), and kidney (μmol/mg) (**r**). **s** and **t** ORO staining with black arrows indicating ORO-positive areas (**s**) and quantification of ORO-positive areas (**t**). **u** The mRNA levels of Lxrβ-driven lipogenic genes. Data are expressed as mean ± SD. Statistical significances were tested by the unpaired two-sided Student's t-test (**a**, **c–e**, **g**, **h**, **k**, **m**, **o**, **q**, **r**, **t**, **u**). Source data are provided as a Source Data file.

effect was abolished by *Sult2a1* knockdown, suggesting that reducing BA overload by Sult2a1-catalyzed BA sulfation contributes to restoring hepatic lipids. Besides, *Lxrβ* is essential for maintaining liver lipogenesis by upregulating lipogenic genes[33]. We found that hepatocyte-specific *Acer3* deletion and CER(d18:1/18:1) treatment upregulated the lipogenic genes *Scd1* and *Fasn*, whereas *Lxrβ* knockdown decreased their mRNA expression and hepatic lipids (Fig. S7c). However, other lipogenic-related nuclear receptors, such as *Srebp1* and *Pparα*, were not significantly affected by CER(d18:1/18:1) treatment (Fig. S8). These findings suggest that the Lxrβ-driven upregulation of lipogenic genes and *Sult2a1* cooperate to maintain hepatic lipid content in *Acer3* deficient female mice with CLI. Since impairment of lipogenesis promotes CLI while improving lipogenesis alleviates CLI[48], preserving lipogenesis is also critical for *Acer3* ablation to attenuate CLI.

CERs are bioactive lipids with multiple signaling transductive functions[14]. CER(d18:1/18:1), the substrates of ACER3, exhibited negative correlations with SCSMs in patients with cholestasis (Fig. 1c), and *ACER3* ablation specifically increased CER(d18:1/18:1) (Figs. 4 and 8). To elucidate the role of CER(d18:1/18:1) in CLI, we applied a CLI mice model treated with CER(d18:1/18:1). CER(d18:1/18:1) treatment remarkably increased CER(d18:1/18:1) while causing only minimal increases in CER(d18:1/18:0) and CER(d18:1/20:1) in the mouse liver (Fig. S8a). The increase of CER(d18:1/18:0) and CER(d18:1/20:1) likely reflects the highly dynamic nature of CER metabolism, wherein CER(d18:1/18:1) likely serves as a precursor in the biosynthesis of other sphingolipids (SLs), such as sphingomyelins (SMs), glucosylceramides (GluCERs), and lactosylceramides (LacCERs), which may subsequently give rise to other CER species through reusing the breakdown products of these complex SLs[14]. Meanwhile, the degradation products of CER(d18:1/18:1), including SPH(d18:1) and fatty acid(18:1), may also be reutilized to generate other CERs[14]. Despite this metabolic flexibility, CER(d18:1/18:1) remained the most prominently elevated CER by CER(d18:1/18:1) treatment (Fig. 5a), supporting the model's relevance for studying the role of CER(d18:1/18:1) in CLI. Functionally, CER(d18:1/18:1) treatment activated LXRβ to improve BA detoxification and lipogenesis, resulting in attenuated CLI in female mice and HepG2 cells (Fig. 5, Fig. S12). Furthermore, *Lxrβ* knockdown reduced the protective effects of CER(d18:1/18:1) treatment against CLI (Fig. 6). These findings uncover that CER(d18:1/18:1) mediates the protective effects of *ACER3* ablation against CLI by activating LXRβ signaling in hepatocytes. Interestingly, *Acer3* ablation activated Lxrβ without affecting its known ligands[34], including OS and certain CHOL derivates (Fig. S9). The nuclear co-localization of CER(d18:1/18:1) and activated LXRβ implicated a direct interaction of CER(d18:1/18:1) and LXRβ. Importantly, SPR analysis and immunoprecipitation validated their interaction (Fig. 8). Our in silico analysis further demonstrated that CER(d18:1/18:1) interacts with key amino acid residues in LXRβ LBD, which are essential for lipid agonists to activate LXRs[40–42]. Interestingly, our docking modeling results further revealed that CER(d18:1/18:1) had a higher binding affinity for LXRβ than for its isotype, LXRα (Table S3, S4), indicating a preference for LXRβ Although treatment of CER(d18:1/18:1) might lead to minimal increases in other CER species, our immunoprecipitation experiments confirmed that CER(d18:1/18:1) was the most abundant CER bound to LXRβ (Fig. S13b), consolidating that

CER(d18:1/18:1) serves as a ligand of LXRβ. However, the precise mechanisms by which ACER3 and CER(d18:1/18:1) modulate LXRβ expression and activity remain to be fully elucidated. Notably, LXR ligands have been shown to suppress the ubiquitination and degradation of LXRs or directly stabilize LXRs[50,51]. Thus, CER(d18:1/18:1) may increase LXRβ protein levels by conferring ligand-mediated post-transcriptional stabilization. Furthermore, previous studies have identified LXR response elements in the promoter region of the LXR genes, where the LXR/RXR heterodimer can bind and activate LXR expression, thereby forming a positive feedback loop[52,53]. Interestingly, our findings demonstrated that *Acer3* knockout also led to the upregulation of Rxrα levels (Fig. S6j, S6k), suggesting that targeting the ACER3-CER(d18:1/18:1) metabolic axis may facilitate the formation of the LXRβ/RXRα heterodimer, which in turn upregulates mRNA expression of LXRβ via a positive feedback mechanism. Overall, our findings indicate that CER(d18:1/18:1) functions as an endogenous agonist of LXRβ, mediating the protective effects of ACER3 ablation against CLI. This provides a molecular insight into the therapeutic potential of CER(d18:1/18:1) for CLI.

Unlike in female mice, *Acer3* deletion only marginally attenuated CLI in male mice, with a slight reduction in hepatic necrosis but no significant effect on inflammation or fibrosis (Fig. S3). Our study also determined whether these sex-specific effects of *Acer3* ablation observed in mice applied to human cholestasis. Mechanistically, we found that Sult2a1 critically mediated the protective function of *Acer3* ablation in female mice, whereas *Acer3* deletion failed to upregulate Sult2a genes in male mice (Fig. S4). Sult2a expression is known to be sex-specific in mice, with higher expression in female livers than males[54]. These findings suggest that the sex-specific expression of Sult2a genes determines the sex-specific protective effect of *Acer3* ablation against CLI in mice. Regarding species differences, Sult2a genes exhibit sex-specific expression in various rodents, with higher expression in female livers of FVB mice, C57BL/6 mice, and Fischer F-344 rats[55–57]. However, this sex-specific expression is absent in guinea pigs and hamsters[58], indicating species-specific variability in Sult2a regulation. In humans, the liver primarily expresses *SULT2A1*[59]. Although sex-specific differences in *SULT2A1* expression have not been reported yet, sulfation activities in the human liver appear similar between females and males[60]. Consistent with this finding, our findings demonstrated that *SULT2A1* expression levels were comparable between men and women, with cholestasis impairing SULT2A1 expression similarly in both sexes (Fig. 7). These results suggest that while Sult2a1 plays a sex-specific role in rodents, this distinction does not extend to humans. In terms of sex-specific differences in CER metabolism, the inherent mRNA levels of CER-metabolizing enzymes showed no significant differences between sexes under either normal conditions or cholestasis (Fig. 7j). However, the changes in the mRNA levels of certain CER-metabolizing enzymes induced by cholestasis varied between sexes (Fig. S10a). Despite these differences, most hepatic CER(d18:1/18:1) species were increased by cholestasis and did not differ significantly between sexes (Fig. 7). Similarly, cholestasis significantly decreased SPH levels in both male and female patients without affecting S1P levels, and SPH and S1P levels showed no significant differences between sexes in either non-CLI or CLI livers

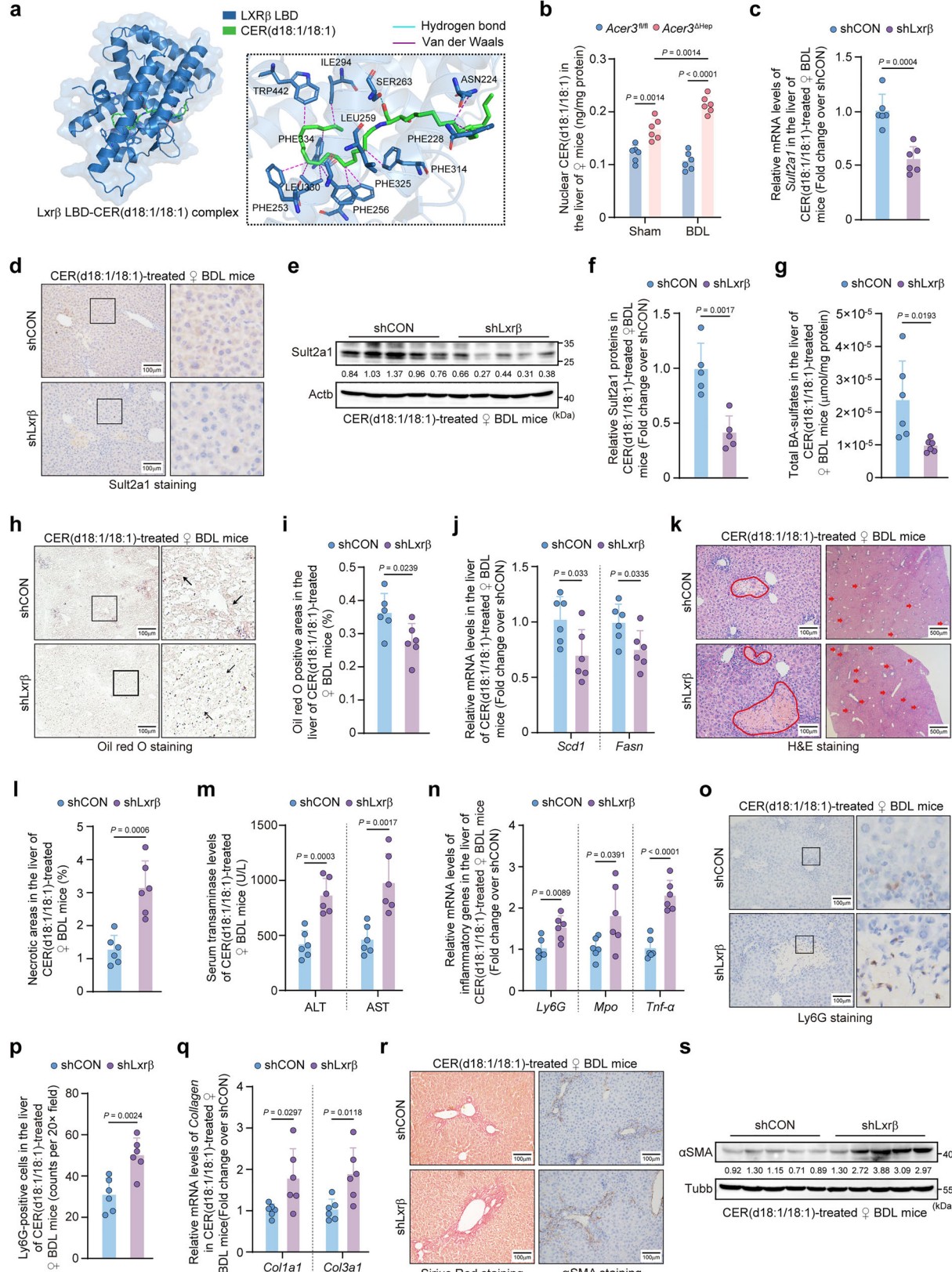

(Fig. S10). These findings suggest that, while cholestasis-induced changes in certain CER-metabolizing enzymes may involve subtle sex-specific regulation, the overall patterns of CER metabolism and activation of CER production are unlikely to be strongly influenced by sex. For instance, although *DEGS2* and *GLA* were found to be differentially upregulated by cholestasis between sexes, both enzymes are involved

in CER generation[61,62], which may contribute to the consistent increase in hepatic CER levels regardless of sex. Importantly, no significant sex-specific differences were observed in *ACER3* expression in human livers, and cholestasis-induced upregulation of *ACER3* and CER(d18:1/18:1) was comparable in both sexes (Fig. 7). These findings indicate that the regulation of *ACER3* and CER(d18:1/18:1) are not sex-specific in

**Fig. 6 | CER(d18:1/18:1) upregulates Sult2a1-catalyzed BA sulfation to attenuate CLI likely by binding Lxrβ. a** Virtual model of Lxrβ LBD-CER(d18:1/18:1) complex. (**b**) Nuclear CER(d18:1/18:1) in the liver of $Acer3^{fl/fl}$ and $Acer3^{\Delta Hep}$ female mice with or without BDL ($n = 6$). **c–g** Sult2a1-catalyzed BA sulfation in the liver of CER(d18:1/18:1)-treated C57BL/6 J WT female BDL mice with or without $Lxrβ$ knockdown ($n = 6$). $Sult2a1$ mRNA levels (**c**), Sult2a1 staining (**d**), Sult2a1 immunoblot (**e**), and quantification of Sult2a1 proteins (**f**) in the liver. Total BA-sulfates in the liver (**g**). **h–j** Lipid content and mRNA expression of lipogenic genes in the liver of CER(d18:1/18:1)-treated C57BL/6 J WT female BDL mice with or without $Lxrβ$ knockdown (n = 6). ORO staining with black arrows indicating ORO-positive areas (**h**) and

quantification of ORO-positive areas (**i**). The mRNA levels of $Scd1$ and $Fasn$ (**j**). **k–s** Examination of CLI in CER(d18:1/18:1)-treated C57BL/6 J WT female BDL mice with or without $Lxrβ$ knockdown ($n = 6$). H&E staining with the circle areas and red arrows indicating necrotic foci (**k**) and quantification of necrotic areas (**l**). Serum transaminase levels (**m**). Inflammatory gene mRNA levels (**n**). Ly6G staining (**o**) and quantification of Ly6G-positive cells (**p**). Collagen mRNA levels (**q**). Sirius Red staining (left panel) and αSMA staining (right panel) (**r**). αSMA immunoblot (**s**). Data are expressed as mean ± SD. Statistical significances were tested by the one-way ANOVA with Tukey's multiple comparisons (**b**) and unpaired two-sided Student's t-test (**c**, **f**, **g**, **i**, **j**, **l-n**, **p**, **q**). Source data are provided as a Source Data file.

humans under normal conditions or cholestasis. Moreover, $ACER3$ knockdown effectively reduced BA toxicity by upregulating $SULT2A1$ in HepG2 cells (Fig. 8), which were originally derived from a male patient[38]. Collectively, our results indicate that the sex-specific expression patterns and functions for $Sult2a1$ and Acer3-mediated CER metabolism observed in mice are unlikely expected in humans, suggesting that targeting $ACER3$ could be an effective therapeutic strategy for CLI in both male and female patients.

While our study highlights the specific role of ACER3-catalyzed CER(d18:1/18:1) hydrolysis in mitigating BA overload in CLI, the broader landscape of CER metabolism in CLI remains underexplored. In particular, special attention may be given to enzymes such as ASAH1, B4GALT6, and DEGS2, along with their substrates, as these were found to be dysregulated by CLI in our study (Fig. 1). For instance, B4GALT6 is involved in the synthesis of LacCERs and has been implicated in inflammatory processes[63,64]. Our ongoing study found that B4galt6, LacCERs, and GluCERs were elevated in mice with cholestasis. Pharmacological inhibition of B4galt6, using D-$threo$-1-phenyl-2-decanoy-lamino-3-morpholino-1-propanol (D-PDMP)[64], reduced inflammation and fibrosis without affecting $Sult2a1$ expression in female mice (Fig. S14). These findings suggest that CER metabolism is tightly regulated by cholestasis, with enzymes such as ACER3, B4GALT6, and others playing critical roles in influencing CLI through distinct mechanisms. Furthermore, our findings demonstrated the sex-specific response to $Acer3$ ablation in mice, which may not extend to humans. Large-scale studies are still needed to validate these observations and provide more robust insights into potential sex-specific differences in CER metabolism, particularly regarding specific CER-metabolizing enzymes. Future studies should pay particular attention to sex differences for CER metabolism, examining whether the sex-specific effects observed in animal models also apply to humans. Lastly, building on the previous study showing signaling transduction regulated by BA influences CER metabolism[20], our study reveals the role of CER in regulating BA metabolism, underscoring the importance of exploring the regulatory interactions between CER and other metabolic pathways to uncover broader pathophysiological roles of metabolic cross-regulation in liver diseases.

In conclusion, our study revealed that $ACER3$ plays a pathological role in CLI by impeding the buildup of CER(d18:1/18:1), while CER(d18:1/18:1) acts as an endogenous agonist of LXRβ to improve BA detoxification and lipogenesis in the liver with CLI. Our work lays the groundwork for future therapeutic interventions targeting $ACER3$ or supplementing CER(d18:1/18:1) to treat cholestatic liver diseases.

## Methods
### Ethics statement
The experiments using human samples were approved by the Medical Ethics Committee of Nanfang Hospital of Southern Medical University under ethical ID NFEC-2021-356. All research was conducted in accordance with relevant guidelines and regulations, and written informed consent was obtained from all patients. The animal experimental procedures were approved by the Institutional Animal Care and Use Committee of Southern Medical University.

### Human samples
Liver tissues were obtained from 30 patients with CLI caused by bile duct obstruction (CLI group) and 30 patients without CLI (non-CLI group) who underwent hepatectomy at the Division of Hepatobilio-pancreatic Surgery, Department of General Surgery, Nanfang Hospital, Southern Medical University (Guangzhou, Guangdong, China) between August 2021 and June 2023. The patient's characters are illustrated in Tables S1 and S2.

### Animal study
Mice with C57BL/6 J genetic background were bred and reared under specific-pathogen-free (SPF) conditions with a 12 h/12 h light/dark cycle at 21 °C and 50-55% humidity at the animal facilities of Southern Medical University. Global $Acer3$ deficient mice ($Acer3^{-/-}$) and wildtype (WT) littermate controls ($Acer3^{+/+}$) were generated as in our previous study[23]. In $Acer3^{-/-}$ mice, the exon 8 of the $Acer3$ gene was replaced by the neomycin-resistant gene cassette[23]. Hepatocyte-specific $Acer3$ deficient mice ($Acer3^{\Delta Hep}$) and littermate controls ($Acer3^{fl/fl}$) were generated by CRISPR/Cas-mediated genome engineering (Cyagen Biosciences Inc, Suzhou, Jiangsu, China). As shown in Figure S2k, exons 3 and 4 of the $Acer3$ gene were selected as conditional knockout regions (cKO region). Homologous arms and cKO region were generated by PCR to engineer the targeting vector. The gRNA targeting the exons 3 and 4, the donor vector containing loxP sites, and Cas9 mRNA were co-injected into fertilized mouse eggs to generate targeted conditional knockout offspring (F0). The F0 mice were bred with mice-expressing hepatocyte-specific Cre recombinase driven by the albumin ($Alb$) gene promoter ($Alb$-cre) to generate F1 mice. The F1 heterozygous mice were bred to generate homozygous $Acer3^{\Delta Hep}$ and $Acer3^{fl/fl}$ mice. The identifications of genotypes are shown in Figure S2l. The gRNA sequences are listed in Table S5. $Sult2a1$ and $Lxrβ$ knockdown were performed with liver-directed type 8 adeno-associated viruses (AAV) carrying shRNA targeting $Sult2a1$ and $Lxrβ$ genes (shSult2a1 and shLxrβ) via tail vein injection, respectively. Corresponding control shRNA (shCON) with a green fluorescent protein (GFP) (Obio Biology, Shanghai, China) was used as control mice. The targeted sequences of vectors in the $Sult2a1$ or $Lxrβ$ genes are listed in Table S6. C57BL/6 J female mice were intraperitoneally injected with 5 mg/kg·day CER(d18:1/18:1) (Avanti Polar Lipids, Birmingham, Alabama, USA) dissolved in 0.5% sodium carboxymethylcellulose (CMC-Na)[65] (Sigma-Aldrich, MO, USA), and 0.5% CMC-Na was used as vehicle control. CER(d18:1/18:1) treatment was performed once every day for 10 days, and BDL was performed on the third day.

BDL was conducted to induce CLI in eight-week-old mice, while Sham operation was used as control. The procedures of BDL were performed with midventral laparotomy and the isolation of the common bile duct above the duodenum, then the common bile duct was ligated at two sites to induce obstructive cholestasis[66] (Figure S2a). Mice were sacrificed 7 days after the BDL operation. The tissues and serum were collected and stored at -80 °C. Liver tissues were fixed in 4% paraformaldehyde (PFA) (Sigma-Aldrich, MO, USA) or Tissue-Tek OCT compound (Sakura Finetek, CA, USA) for pathophysiological examination.

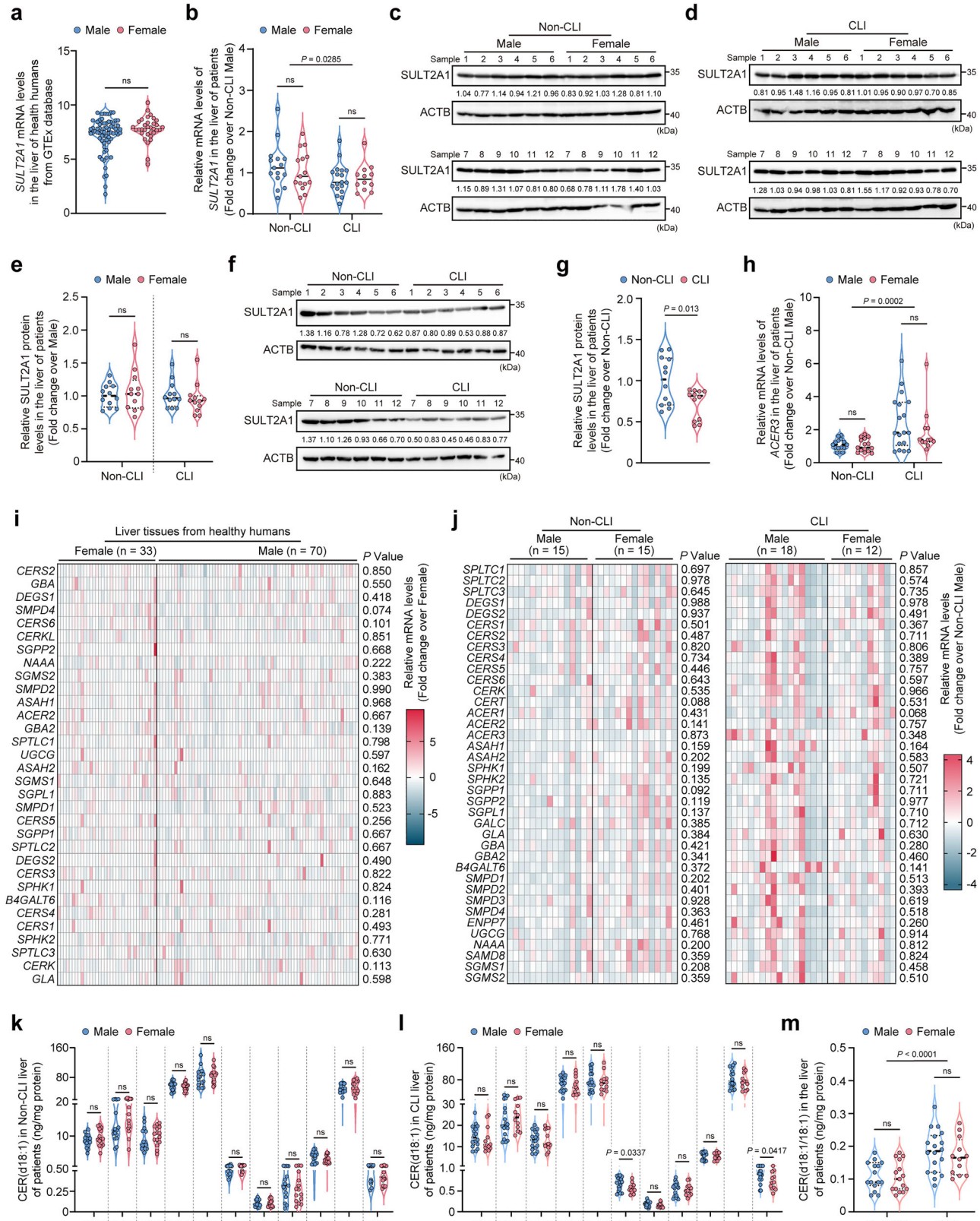

## Cell culture and experimental conditions

The human-liver-derived cell lines HepG2, Huh-7, Hep3B, and MHCC97-H were obtained from the Shanghai Cell Bank of the Academy of Chinese Sciences and Liver Cancer Institute (Zhongshan Hospital, Fudan University, China). These cell lines were authenticated through short tandem repeat analysis to verify their identity and uniqueness. These cell lines were cultured in the Dulbecco's modified Eagle's medium (DMEM, Gibco, USA) or Roswell Park Memorial Institute medium 1640 (RPMI, Gibco, USA) supplemented with 10% fetal bovine serum (FBS, Gibco, USA) and 1% Penicillin/Streptomycin (P/S, Gibco, USA) in a humidified incubator at $37\,^{\circ}\text{C}$ with 5% $CO_2$. For mimicking CLI, HepG2 cells were treated with lithocholic acid (LCA, Sigma-Aldrich, MO, USA) at concentrations of 100 and 200 μM and 0.1% DMSO (Sigma-Aldrich, MO, USA) vehicle control for 12 hours, and

**Fig. 7 | Sex has minimal impact on the expression of *SULT2A1* and CER-related enzymes and CER(d18:1/18:1) in the human liver. a** *SULT2A1* mRNA levels in the healthy liver tissues of male (*n* = 70) and female (*n* = 33) humans from the Genotype-tissue expression (GTEx) database. **b**–**g** Expression levels of *SULT2A1* in the collected liver tissues of male and female patients with or without cholestasis. *SULT2A1* mRNA levels (**b**). Immunoblot of SULT2A1 in the collected liver tissues of male (*n* = 12) and female (*n* = 12) patients with non-CLI (**c**) or CLI (**d**) and quantification of SULT2A1 expressions (**e**). Immunoblot of SULT2A1 (**f**) in the collected liver tissues of patients with non-CLI (*n* = 12) or CLI (*n* = 12) and quantification of SULT2A1 expressions (**g**). **h** *ACER3* mRNA levels in the collected liver tissues of male

and female patients. **i** Heat map of the mRNA levels of CER-metabolizing enzymes in the healthy liver tissues of male (*n* = 70) and female (*n* = 33) humans from the GTEx database. **j** Heat maps of the mRNA levels of CER-metabolizing enzymes in the collected liver tissues of male and female patients with non-CLI (left panel) and CLI (right panel). **k**–**m** CER(d18:1) levels in the collected liver tissues of male and female patients with non-CLI (**k**) and CLI (**l**). CER(d18:1/18:1) levels (**m**). Data are expressed as mean ± SD. Statistical significances were tested by the unpaired two-sided Student's t-test (**a**, **g**, **k**, **l**) and one-way ANOVA with Tukey's multiple comparisons (**b**, **e**, **h**, **m**). Source data are provided as a Source Data file.

then the cells and conditional medium were harvested to assess the expression of apoptosis-associated proteins and LCA-sulfate. For *ACER3* knockdown, HepG2 cells were transfected lentivirus-carried plasmids of short-hairpin RNA (shRNA) targeting *ACER3* (shACER3) or vector control (shCON) (Jikai Biology, Shanghai, China) (MOI = 10) for 72 hours, and stable *ACER3*-knockdown HepG2 cells were obtained with puromycin-resistance-based (2 µg/ml) screening. The *ACER3*-knockdown HepG2 cells were transfected with SULT2A1 or LXRβ siRNA (siSULT2A1 or siLXRβ) and their respective control siRNA (siCON) (Jiyuan Biology, Shanghai, China) using RNA iMAX transfection reagent (Invitrogen, California, USA) following the manufacturer's instructions. For CER(d18:1/18:1) treatment, HepG2 cells were pretreated by minimal essential medium (MEM, Gibco, USA) for 12 hours, and then incubated with 5 µM CER(d18:1/18:1) or vehicle (2% dodecane dissolved in ethyl alcohol) (Sigma-Aldrich, MO, USA) for 24 hours, followed by incubation with 200 µM LCA or vehicle (0.1% DMSO) for 12 hours. In SULT2A1-, LXRβ-, and LXRα-siRNA transfected HepG2 cells, siRNA transfection was performed for 24 hours followed by treatment with 5 µM CER(d18:1/18:1) or vehicle (2% dodecane) for another 24 hours, then added 200 µM LCA as described above. All in vitro results presented in this study represent at least three independent experiments. The sequences of indicated shRNA and siRNA are provided in Table S7.

### GTEx database analysis
RNA sequencing data including 70 male and 33 female liver tissues were obtained from the GTEx database (https://www.gtexportal.org/) and used for analysis in this study. R version 4.3.2 was used to conduct a gene expression analysis of CER-metabolizing genes between female and male livers and corresponding visualization.

### Transcription factor prediction
ACER3-related transcription factor prediction was accomplished by the online predicted tool (https://jingle.shinyapps.io/TF_Target_Finder/). Utilizing the hTFtarget, ChIP-Atlas, GTRD, ENCODE, and JASPAR databases, all of the ACER3-related potential transcription factors from different databases were selected and visualized in the form of a Venn chart.

### Luciferase activity assay
The promoter of SULT2A1 (-2000 to -1 bp) was subcloned and inserted into a pGL3-basic vector (Promega) (Figure S11j). The *ACER3*-knockdown HepG2 cells and *ACER3*-knockdown HepG2 cells transfected with LXRβ siRNA were further transfected with pGL3-SULT2A1-luc or pGL3-basic vector plasmid using Lipofectamine 3000 (Invitrogen, CA, USA). pRL-TK was transfected to normalize the efficiency of transfection. Luciferase receptor assays were performed using a Dual-luciferase assay kit (Promega, Madison, USA) 24 hours after transfection. The luciferase activity was determined by the Gen5 (Biotek, Washington, USA). All reporter assays were repeated three times.

### Liver in situ hybridization (ISH)
Formalin-fixed and paraffin-embedded (FFPE) slides of mouse liver tissues were subjected to ISH using RNA scope ISH kits and probes (Advanced Cell Diagnostics, CA, USA) as described by Wang F[67]. Liver

sections were pretreated by repair reagents and then hybridized with the specific oligonucleotide probe targeting the region (93-1195 bp) of the *ACER3/Acer3* gene. After amplification of the staining signal, sections were hybridized with a probe labeled with horseradish peroxidase (HRP). Positive staining was detected with a red color. Each RNA transcript exhibited a distinct dot or cluster of signals.

### Liver histopathological examination
FFPE slides of mouse liver tissues were subjected to hematoxylin and eosin (H&E) staining for histopathological examination. The assessment of collagen formation was assessed by staining using commercial Sirius red dye (Solarbio, Beijing, China). Liver injury was evaluated by detecting the levels of alanine aminotransferase (ALT) and aspartate aminotransferase (AST) in the mouse serum using ALT and AST Colorimetric Activity Assay Kits (Sigma-Aldrich, MO, USA).

### Oil Red O staining
Oil Red O staining was performed to assess the lipid content in the liver. Liver frozen sections or cells were fixed and then incubated with Oil Red O solution (0.375%, wt/vol) (Sigma-Aldrich, MO, USA) for 5 min. The samples were immersed in ddH$_2$O for 1 min, mounted using a water-soluble mounting medium, and examined under the Intelligently Designed Microscope (Olympus, Shinjuku-ku, Tokyo, Japan).

### Immunohistochemistry (IHC)
IHC staining was performed using a VECTASTAIN® Elite® ABC Kit (Rabbit IgG) (VECTOR, Burlingame, CA, USA) and DAB Peroxidase Substrate Kit (VECTOR, Burlingame, CA, USA) following the manufacturer's instructions. Liver sections were subjected to IHC staining with antibodies against ACER3 (Sigma-Aldrich, MO, USA), Alpha-smooth muscle actin (αSMA) (Cell Signaling Technology, Danvers, MA, USA), SULT2A1 (Abcam, Cambridge, MA, USA), lymphocyte antigen 6 complex locus G6D (LY6G) (Abcam, Cambridge, MA, USA).

### Image analysis
Necrotic and Oil Red O positive areas were quantified by analyzing 5 randomly selected fields (20 ×) per section using Image Pro Plus software (Media Cybernetics, Bethesda, MD, USA). The assessment of Ly6G and ACER3 staining was performed by counting positively stained cells in 5 randomly selected fields per section within a 20 × field of view using a blind approach. Representative pictures were taken using the Intelligently Designed Microscope (Olympus, Shinjuku-ku, Tokyo, Japan).

### Immunofluorescence (IF) assay
IF co-staining was performed on the liver frozen sections and HepG2 cells using LXRβ antibody (Abcam, Cambridge, MA, USA) and ALB antibody (Proteintech, Guangzhou, China). Alexa Fluor® 488-conjugated rabbit antibody (Abcam, Cambridge, MA, USA) and Alexa Fluor® 594-conjugated mouse antibody (Abcam, Cambridge, MA, USA) were used as secondary antibodies. The cell nucleus was counterstained with DAPI (Abcam, Cambridge, MA, USA). The stained liver sections were analyzed using the Intelligently Designed Microscope (Olympus, Shinjuku-ku, Tokyo, Japan). The confocal dishes were

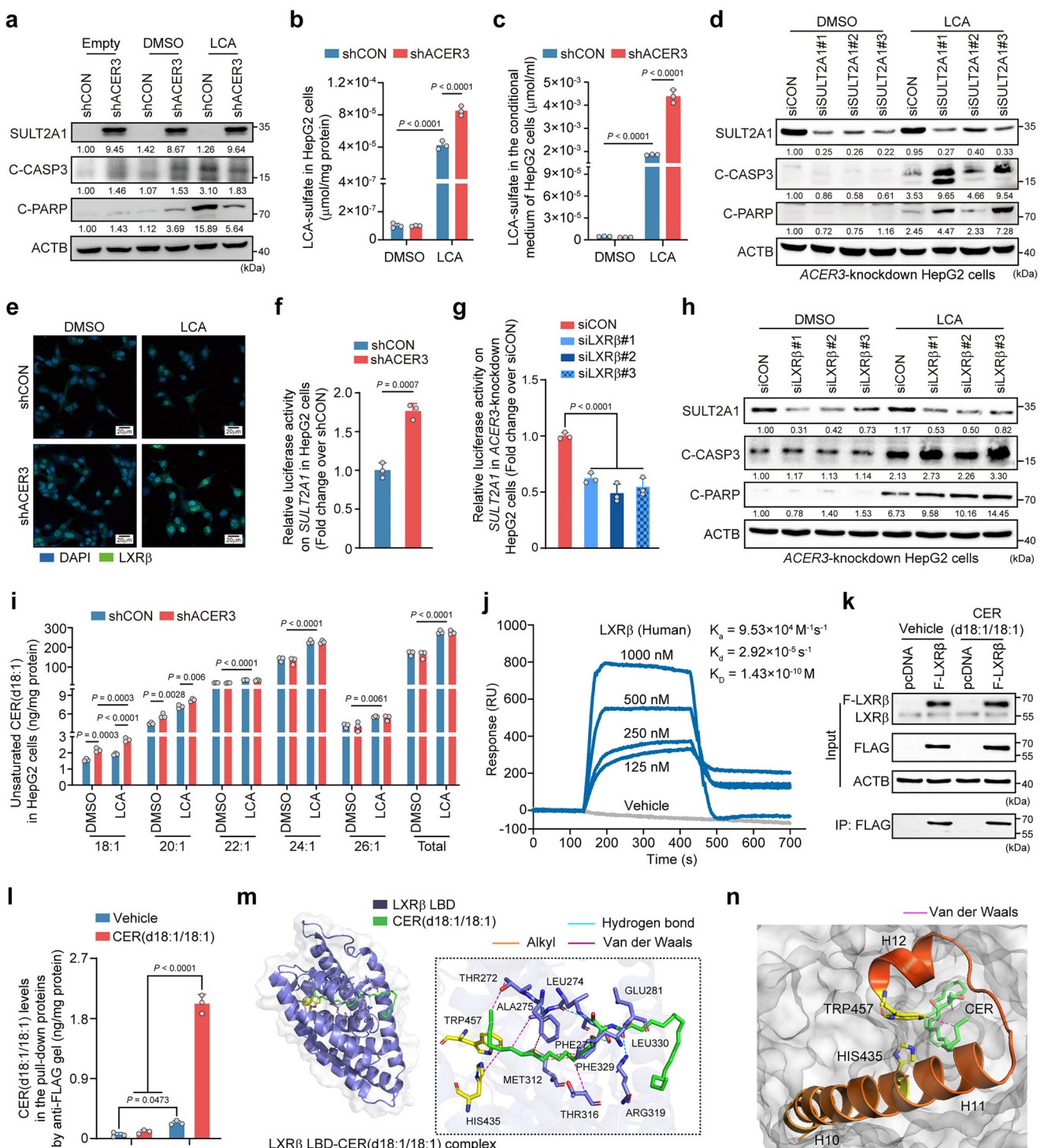

**Fig. 8 | Targeting *ACER3* attenuates CLI by upregulating *SULT2A1* through the CER(d18:1/18:1)-LXRβ interaction in HepG2 cells. a** Immunoblots of SULT2A1, cleaved-caspase 3 (C-CASP3), and cleaved-poly ADP-ribose polymerase (C-PARP) in HepG2 cells transfected by shCON and shACER3 lentivirus following treatment of vehicle (transfection medium), dimethyl sulfoxide (DMSO), or 200 μM lithocholic acid (LCA). **b** and **c** LCA-sulfate in HepG2 cells (**b**) and the conditional medium (**c**) with or without *ACER3* knockdown and LCA treatment. (**d**) Immunoblots of SULT2A1, C-CASP3, and C-PARP in *ACER3*- and *SULT2A1*-knockdown HepG2 cells with or without 200 μM LCA. **e** and **f** LXRβ immunofluorescence (**e**) and luciferase activity on SULT2A1 promoter (**f**) in shCON and shACER3 HepG2 cells with or without LCA. **g** and (**h**) Luciferase activity on SULT2A1 promoter (**g**) and immunoblots of SULT2A1, C-CASP3, and C-PARP (**h**) in *ACER3*-knockdown HepG2 cells

with or without *LXRβ* knockdown. (**i**) Unsaturated CER(d18:1) in shCON and shACER3 HepG2 cells with or without LCA treatment. **j** Surface plasmon resonance (SPR) titration curves reflect the interaction between recombinant human LXRβ and CER(d18:1/18:1). **k** Flag-LXRβ immunoprecipitation. **l** CER(d18:1/18:1) in the immunoprecipitated LXRβ-FLAG proteins. **m** Virtual structure of human LXRβ ligand-binding domain (LBD)-CER(d18:1/18:1) complex. **n** Interaction between CER(d18:1/18:1) and the agonism-related residues within LXRβ LBD. Images and results represent the results of three independent experiments. Data are expressed as mean ± SD. Statistical significances were tested by the one-way ANOVA with Tukey's multiple comparisons (**b**, **c**, **g**, **i**, **l**) and unpaired two-sided Student's t-test (**f**). Source data are provided as a Source Data file.

observed and analyzed using a Laser Scanning Confocal Microscope LSM 980 (ZEISS, Oberkochen, Germany). Nuclear LXRβ/Lxrβ-positive cells were quantified within 5 randomly selected fields (20 ×) per section using Image Pro Plus software.

## Immunoprecipitation (IP)

The Flag-tagged human LXRβ (F-LXRβ) coding sequence was constructed from the respective cDNA clones using 3 × FLAG-tag-encoding oligonucleotides, followed by insertion into the pcDNA$^{TM}$3.1 vector (Thermo Fisher Scientific, Waltham, MA, USA). HepG2 cells were transfected with plasmids containing F-LXRβ or empty vectors. After 24-hour transfection, the cells were treated with CER(d18:1/18:1) for another 24 hours. The proteins were extracted using a detergent-free Minute$^{TM}$ Total Protein Extraction Kit (Invent Biotechnologies, Eden Prairie, USA). The expression efficiency of FLAG and LXRβ protein was verified by Western blot. EZview Red ANTI-FLAG M2 Affinity Gel (Sigma-Aldrich, MO, USA) was washed with Tris Buffered Saline solution (50 mM Tris HCl, 150 mM NaCl, pH 7.4) twice. Diluted protein lysates (100 ul) were incubated with EZview Red ANTI-FLAG M2 Affinity Gel (20 ul) and shaken slowly at 4 °C overnight. The next day, the samples were centrifuged at 8200 g for 30 s at 4 °C and the supernatant was removed. The precipitates were gently mixed and incubated with Tris-buffered saline solution for 5 min, centrifuged at 8200 g for 30 s, and the supernatant was removed. Repeat the washing step three times. The above precipitations were incubated with 150 μl FLAG peptide (Sigma-Aldrich, MO, USA) at 4 °C for 30 min. The suspension was centrifuged at 4 °C, 8200 g for 30 s. The supernatants were verified for the efficiency of IP and then subjected to CER extraction for CER measurement.

## RNA isolation and quantitative real-time polymerase chain reaction (RT-qPCR) analysis

Total RNA was extracted from human liver tissues, mouse liver tissues, and cells, using TriZol reagent (Invitrogen, Waltham, MA, USA). Subsequently, mRNA was transcribed into cDNA by 5 × PrimeScript RT Master Mix (TaKaRa, Kusatsu, Shiga, Japan). RT-qPCR analyses were done on LightCycler 480 (Roche, Auckland, New Zealand). Relative gene expression levels were determined using ΔCT calculation, and mRNA levels were relative to the control condition where indicated. *Actb* was used as the housekeeping control. Primer's sequences of indicated mRNA of human and mouse genes employed for the RT-qPCR are illustrated in Table S8.

## Western blotting

Liver tissues and cell pellets were homogenized in RIPA buffer (Thermo Scientific, Waltham, MA, USA) to extract whole-cell protein. Experiments with the separation of nuclear and cytoplasmic proteins were performed using a Nuclear and Cytoplasmic Protein Extraction Kit (Invent Biotechnologies, Eden Prairie, USA) according to the manufacturer's instructions. The concentrations of protein extracts were determined using a bicinchoninic acid (BCA) protein determination kit (Thermo Fisher Scientific, Waltham, MA, USA). An equal amount of denatured protein with loading buffer was loaded in each lane of the 10% or 12% SDS-PAGE Tris-glycine gels and transferred onto polyvinylidene fluoride (PVDF) membranes (Roche, Auckland, New Zealand) using electrophoretic wet Western blot transfer system. After blocking, the membranes were incubated with primary antibodies overnight followed by incubation with HRP-conjugated secondary antibodies. The signals were visualized using the ECL Prime Western Blotting Detection Reagent (Cytiva, Westboro, USA). ACTB/Actb, TUBB/Tubb, and GAPDH/Gaphd were used as the loading control for the whole-cell samples. GAPDH/Gaphd and H3F3A/H3f3a were used as loading controls for the nuclear and cytoplasmic proteins, respectively. The information on the used antibodies is listed in Table S9.

## RNA sequencing (RNA-seq)

Total RNA was extracted from the liver tissues of *Acer3*$^{+/+}$ and *Acer3*$^{-/-}$ mice. Illumina RNA-seq libraries were prepared by Shanghai Majorbio Bio-Pharm Technology Co., Ltd. The libraries were sequenced on an Illumina Novaseq 6000 platform. The mouse genomic and genetic information was obtained from the National Center for Biotechnology Information database. Expression levels of mRNA were evaluated using StringTie software (v1.3.44 d). For data analysis, |log2FC| ≥1 and *P*-value ≤ 0.05 were considered as the threshold criteria to screen differentially expressed genes (DEGs). The obtained data were used to generate fold changes and transform them to draw volcano plots.

## Bile acids (BAs) measurement

Liver tissues were homogenized in 40% methanol (MeOH). Subsequently, the mixtures were incubated at 4°C for 30 min. Samples were centrifuged at 1500 g at the end of incubation and clean supernatant was extracted. The extraction was repeated with ice-cold MeOH: chloroform (3:1) and the extracts were pooled. Pooled extracts were dried in a SpeedVac under OH mode, and resuspended in 50 μL MeOH containing deuterated internal standards (IS) before liquid chromatography-mass spectrometry (LC-MS) analysis. The IS cocktail contained glycochenodeoxycholic acid-d4, glycocholic acid-d4, glycodeoxycholic acid-d4, cholic acid-d4, ursodeoxycholic acid-d4, chenodeoxycholic acid-d4, deoxycholic acid-d4, and lithocholic acid-d4 (Avanti Polar Lipids, Birmingham, Alabama, USA). BAs were determined on an Exion AD30-UPLC coupled with Sciex QTRAP 6500 Plus under electrospray ionization mode. Individual BAs were separated on a Phenomenex Kinetex C18 column (100 × 2.1 mm, 1.7 μm) using 2% formic acid in water as mobile phase A and acetonitrile:isopropanol (1:1) as mobile phase B, and quantitated by referencing to the intensities of their corresponding deuterated IS[68,69]. Measurement of BA-sulfates was performed following the methodology described by Jiangeng Huang[68]. Approximately 100 mg tissues or 40 mg cell pellets were homogenized in two times volumes of ddH$_2$O. Cell culture medium was collected, frozen, and gasified on a vacuum freeze dryer, and the powder was redissolved by ddH$_2$O. The Samples for the extract of BA-sulfates include the homogenate of tissues or cell pellets, redissolved-liquid of cell culture medium and mouse serum. The samples (100 μl) were spiked with 10 μl IS (glycochenodeoxycholic acid-d4) and 2 ml ice-cold alkaline acetonitrile (ACN) (5% NH$_4$OH in ACN) (Sigma-Aldrich, MO, USA). The mixtures were vortexed continuously for 30 min at 4 °C and then centrifuged at 16,000 g for 10 min at 4 °C. The supernatant was aspirated and the precipitation was repeated with 1 ml ice-cold alkaline ACN. The supernatants from the 2 extractionsteps were pooled, evaporated, and reconstituted in 100 μl 50% MeOH. Subsequently, the determination of BA-sulfates was performed by liquid chromatography-tandem mass spectrometry (LC-MS/MS) using prelude SPLC coupled with the TSQ Quantiva system (Thermo Fisher Scientific, Waltham, MA, USA). BA-sulfate standards were obtained from Sigma-Aldrich (MO, USA), including lithocholic acid 3-sulfate (LCA-S), taurolithocholic acid 3-sulfate (T-LCA-S), cholic acid 3-sulfate (CA-S). Taurocholic acid 3-sulfate (T-CA-S), glycodeoxycholic acid 3-sulfate (G-DCA-S), and taurochenodeoxycholic acid 3-sulfate (T-CDCA-S) were obtained from BePure (Beijing, China). The identities of individual BA-sulfates were confirmed based on multiple reaction monitoring transitions and retention times relative to authentic reference compounds. Multiple reaction monitoring parameters of BA-sulfates are provided in Table S11. BA content was normalized to the protein levels of tissues and cells or the volume of serum and cell culture medium.

## Untargeted lipidomics

Lipid extraction from the liver tissues was performed following the Matyash procedure as described in[70]. Equal amounts of liver tissues were used for lipid extraction. The samples were analyzed using

Thermo Fisher Scientific Vanquish Flex ultra-high-performance liquid chromatography (UHPLC) equipped with Thermo Fisher Scientific Orbitrap Fusion Tribrid High-Resolution Mass Spectrometer (Thermo Fisher Scientific, Waltham, MA, USA). The identification of lipid molecular species was conducted by Lipid Search software (Thermo Fisher Scientific, Waltham, MA, USA). Missing values that were not detected in all samples were excluded. $Log_{10}$-transformed data were scaled for principal component analysis (PCA). The differences of individual lipid species were statistically analyzed by Student's *t*-test. The alteration of individual lipid species by BDL or hepatocyte-specific *Acer3* ablation was illustrated by volcano plots.

### Targeted lipidomics

The measurement of CER metabolites was performed following to protocol reported by Wang[71]. In brief, 225 µl MeOH was added to the homogenates of liver tissues, HepG2 cells, and the nucleus of HepG2 cells. After vortexing, 50 µl IS cocktail and 750 µl pre-cooling alkaline methyl tert-butyl ether (MTBE) were added. The mixture was incubated in a Thermomixer Comfort at 650 g for 1 hour at 4 °C. Afterward, 188 µL MilliQ water was added, and the samples were centrifuged at 10,000 g for 10 min at 4 °C. The upper organic layer (600 µl) was transferred to a new tube and dried under a continuous stream of nitrogen to obtain lipid extracts (1 L/min $N_2$ at 25 °C). After removing the middle layer, the lower layer was added with 903 µl MeOH and stored at -80 °C for 4 hours to precipitate protein. Protein pellets were collected after centrifugation at 19,803 g for 30 min at 4 °C and resuspended in buffer solution (1% SDS, 150 mM NaCl, 50 mM Tris, pH 7.8) for protein quantification by BCA. The dried powders from the upper organic layer were resuspended in 100 µl 30% mobile phase B (IPA/ACN, 9/1 (v/v), 0.1 % formic acid, 10 mM ammonium formate, and 5 µM phosphoric acid). CERs were determined by LC-MS/MS performed on prelude SPLC coupled with the TSQ Quantiva system (Thermo Fisher Scientific, Waltham, MA, USA). The identities of individual CERs were confirmed based on multiple reaction monitoring transitions and retention times relative to authentic reference compounds. Multiple reaction monitoring parameters of CERs are provided in Table S10. The IS cocktail contained CER(d18:1/17:0), sphingosine (SPH) (d17:1), sphingosine-1-Phosphate (S1P) (d17:1). CER standards were purchased from Avanti Polar Lipids (Birmingham, Alabama, USA), including CER(d18:1/6:0), CER(d18:1/16:0), CER(d18:1/18:0), CER(d18:1/18:1), CER(d18:1/20:0), CER(d18:1/22:0), CER(d18:1/24:0), CER(d18:1/24:1), SPH(d18:1), and S1P(d18:1). Amounts of sphingolipids were quantified using standard curves and normalized to protein contents.

For targeted lipidomic of oxysterol (OS), lipids were extracted from tissues following Bligh and Dyer's protocol and resuspended in 500 µl ethanol containing 5 µg butylated hydroxytoluene (BHT). The samples were mixed with an IS cocktail (50 µl) comprising d7-24-hydroxcholesterol, d7-7β-hydroxycholesterol, d6-25-hydroxycholesterol, d6-27-hydroxycholesterol, d7-7-keto-cholesterol, d7-7α-hydroxy-cholestenone, d6-TMAS, d7-4β-hydroxycholesterol, d6-24,25-epoxycholesterol, d7-desmosterol, d3-3β-7α-dihydroxycholest-5-enoic acid (Avanti Polar Lipids, Birmingham, Alabama, USA). The mixtures were incubated at 1200 g for 15 min at 4 °C. At the end of incubation, 250 µl MilliQ water and 1 ml n-hexane were added. The samples were mixed thoroughly by vortexing, and centrifuged at 12,000 g for 5 min 4 °C. Clear upper phase containing OS and sterols in hexane was transferred to a new tube. The extraction was repeated once with another 1 ml n-hexane. The pooled extract was dried in a SpeedVac under organic mode. OS were derivatized to obtain their picolinic acid esters before LC/MS analysis on a Thermofisher U3000 DGLC coupled to Sciex QTRAP 6500 Plus/Shimadzu 40X3B-UPLC coupled to Sciex QTRAP 6500 Plus, and quantitated by referencing to the spiked internal standards[72]. Mobile phase A: a gradient system consisting of acetonitrile:methanol:water (45:45:40, v/v/v) with 0.1%

acetic acid. Mobile phase B: acetonitrile:methanol:water (45:45:10, v/v/v) with 0.1% acetic acid. Flow rate: 550 µL/min. Column oven temperature: 35 °C. The gradient program was as follows: 0–2 min, 5% B; 2–3 min, 5–25% B; 3–6 min, 25–65% B; 6–8 min, 65–100% B; 8–12 min, 100% B; 12–13 min, 100–5% B; 13–15 min, 5% B. Amounts of OS were normalized to protein contents.

### Molecular docking

The structure of CER(d18:1/18:1) (Compound CID: 5283563) was downloaded from the PubChem database (https://pubchem.ncbi.nlm.nih.gov/). The crystal structures of the ligand-binding domain (LBD) of human LXRβ protein (UniProt ID: P55055, PDB code: 1p8d, 1upw, 1upv, 1pq9, 3l0e, 3kfc, 4dk8, 4dk7, 4rak, 5hjp, 4nqa, 5i4v, 5kyj, 5 kya, 5jy3, 6k9m, 6k9h, 6k9g, 6s4n, 6s4u, 1pq6, 1pqc, 6jio, 6s4t, and 6s5k) and LXRα protein (UniProt ID: Q13133, PDB code: 2acl, 3fc6, 3fal, 3ipq, 3ips, 3ipu, 5avi, 5avl, and 5hjs) were downloaded from the RCSB Protein Data Bank (http://www.rcsb.org/). All redundant atoms except the chain involved in docking were deleted. The protein structure was treated in several steps including residue repair, protonation, and partial charges assignment in the AMBER ff14SB force field. The DMS tool was employed to build the molecular surface of the receptor using a probe atom with a 1.4 Å radius. The binding pocket was defined by the crystal ligand and spheres were generated filling the site by employing the Sphgen module in UCSF Chimera (version 1.17.3)[73]. Subsequently, the DOCK 6.9[74] program was utilized to execute semi-flexible docking where 10000 different orientations were produced. Clustering analysis was performed (RMSD threshold 2.0 Å) for candidate poses, and the best-scored molecular modeling was output. The crystal structures of LBD of human LXRβ protein by X-ray diffraction with resolution at 2.61 Å (PDB code 5i4v) (https://doi.org/10.2210/pdb5I4V/pdb)[42] were scored with the highest predicted binding affinity to CER(d18:1/18:1) and were selected for docking analysis. The grid scores between human LXRβ or LXRα and CER(d18:1/18:1) were displayed in Tables S3 and S4. Discovery Studio software was used to analyze the hydrophobicity of the ligand-protein interaction.

An online protein structure prediction server, SWISS-MODEL, was used to construct the three-dimensional structure of the LBD of mouse Lxrβ protein (UniProt ID: Q60644). Human LXRβ protein (PDB code: 1pq6, resolution: 2.40 Å) was used as a template through BLAST (identity: 99.58%). Chain A of the protein with the known LXRβ ligand GW3965 was used for model pocket construction. After homology modeling, the molecular docking between CER(d18:1/18:1) and mouse Lxrβ was accomplished as described above.

### Surface plasmon resonance (SPR)

SPR analysis was performed using the PlexArray HT A100 (Plexera; Seattle, USA). Briefly, CER(d18:1/18:1) (Avanti Polar Lipids, Birmingham, Alabama, USA) was loaded on the 3D photo-crosslinking chip, followed by desiccation in a vacuum drier. After the photo-crosslinking reaction, the chip was immersed in Dimethylformamide (DMF) (Sigma-Aldrich, MO, USA), ethanol, and ddH₂O, sequentially. Next, the chip was dried under a continuous stream of nitrogen. The recombinant human LXRβ (AntibodySystem, Schiltigheim, France) (dissolved in PBS) with increasing concentrations (0, 125, 250, 500, 1000 nM) flowed through the chip. The flow rate was set at 2 µl/s. The data was collected with Plexera Data Explorer and analyzed with BIA evaluation software (version 4.1).

### Statistical analysis

Statistical analyses were performed with Statistical Product and Service Solutions software 20.0 (IBM; Armonk, NY, USA) and R version 4.3.2 with ggplot2 and psych packages. Correlation analysis was performed by the Spearman correlation test. Data were means ± SD, analyzed by unpaired two-sided Student's *t*-test or one-way ANOVA

with Tukey's multiple comparisons, and a *P* value < 0.05 (bilateral) was considered significant.

## Reporting summary

Further information on research design is available in the Nature Portfolio Reporting Summary linked to this article.

## Data availability

All data generated or analyzed during this study are included in this paper and its supplementary information files. The RNA-seq data of mouse liver tissues generated in this study have been deposited in the Sequence Read Archive (SRA) database under accession codes PRJNA1182846 and PRJNA1184181. The RNA-seq data of human liver tissues were obtained from GTEx database (https://www.gtexportal.org/). The raw data of the mouse hepatic lipidome have been deposited in the MetaboLights database under accession code MTBLS12198. Utilizing the hTFtarget, ChIP-Atlas, GTRD, ENCODE, and JASPAR databases, transcription factor prediction was accomplished by the online predicted tool (https://jingle.shinyapps.io/TF_Target_Finder/). Source data are provided with this paper.

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

## Acknowledgements

We would like to acknowledge the technical support provided by LipidALL Technologies (Changzhou, Jiangsu, China) and Central Laboratory of Southern Medical University (Guangzhou, Guangdong, China) for lipidomics and bile acid analyses. This work was supported by the National Natural Science Foundation of China (82170647 to K.W, 82270661 to C.J.L, and 82070642 to J.Z), the Basic and Applied Basic Research Foundation of Guangdong Province (2023A1515010088 to C.J.L and 2024A1515013204 to K.W), the Project of Guangzhou Science and Technology (202201020604 and 20231A011030 to L.L), and the Innovation and Entrepreneurship Training Program for College Students (S202212121102 and S202312121240 supervised by K.W).

## Author contributions

Conceptualization, L.Y.L., Z.L., and K.W., Data curation, L.Y.L. and Q.L., Formal analysis, L.Y.L., K.W., Z.L., C.H., Y.L., and Q.L., Investigation, L.Y.L.,

Z.L., and L.L., Validation, L.Y.L., Z.L., L.L., and C.H., Methodology, Z.L., C.H., Y.L., R.X., H.L., C.L., Y.P., X.W., B.W., and Y.X.L., Software, H.B.L., Visualization, L.Y.L., and Z.L., Project administration, L.L., Q.L., and C.J.L., Resources, L.L., T.L., S.Y., J.Q., C.J.L., and K.W., Funding acquisition, L.L., J.Z, C.J.L., and K.W., Supervision, Q.L., C.J.L., and K.W., Writing - original draft, L.Y.L., C.M., and R.X., Writing - review & editing, K.W.

## Competing interests

The authors declare no competing interests.

## Additional information

[1]Division of Hepatobiliopancreatic Surgery, Department of General Surgery, Nanfang Hospital, Southern Medical University, Guangzhou, Guangdong, China. [2]Department of Infectious Diseases, Nanfang Hospital, Southern Medical University, Guangzhou, Guangdong, China. [3]Department of Obstetrics, Guangzhou Women and Children's Medical Center, Guangzhou Medical University, Guangdong Provincial Clinical Research Center for Child Health, Guangzhou, Guangdong, China. [4]Department of Radiation Oncology, Nanfang Hospital, Southern Medical University, Guangzhou, Guangdong, China. [5]Department of Medicine and Cancer Center, The State University of New York at Stony Brook, Stony Brook, New York, USA. [6]Central Laboratory, Southern Medical University, Guangzhou, Guangdong, China. [7]Prenatal Diagnostic Center, Guangzhou Women and Children's Medical Center, Guangzhou Medical University, Guangdong Provincial Clinical Research Center for Child Health, Guangzhou, Guangdong, China. [8]These authors contributed equally: Leyi Liao, Ziying Liu, Lei Liu. ✉e-mail: liqingping1993@smu.edu.cn; licj@smu.edu.cn; kaiwang@smu.edu.cn

