## [Peer Review File · Nature Communications]

Targeting the Ceramidase ACER3 Attenuates Cholestasis in Mice by Mitigating Bile Acid Overload via Unsaturated Ceramide-mediated LXR β Signaling Transduction

Corresponding Author: Dr Kai Wang

Version 0:

Reviewer comments:

Reviewer #1

(Remarks to the Author)

The manuscript titled "Targeting ACER3 Attenuates Cholestasis by Mitigating Bile Acid Overload via Unsaturated Ceramide-mediated LXR β Signaling Transduction" significantly contributes to our understanding of the role of ceramide metabolism in cholestatic liver injury (CLI). The study explores the pathological role of ACER3 and its substrate CER(d18:1/18:1) in CLI, revealing that the ablation of ACER3 in female mice ameliorates CLI through the LXR β -SULT2A1 axis. The findings suggest potential therapeutic targets for CLI, particularly through the modulation of ceramide metabolism. However, several aspects of the study require further attention and clarification.

Major Comments:

- 1. Sex Differences in ACER3 Function:** The study identifies a significant sex difference in the effects of ACER3 ablation, with protective effects observed only in female mice. This difference is attributed to the higher basal expression of Sult2a1 in females. However, the manuscript does not fully explore the implications of this sex difference for potential human therapies. The authors should provide additional data and discussion on SULT2A1 expression levels in human liver samples, particularly regarding potential sex differences. This would strengthen the relevance of the findings for clinical applications.
- 2. Mechanistic Insights and Molecular Interactions:** The manuscript discusses CER(d18:1/18:1) as an agonist of LXR β , leading to the upregulation of SULT2A1 and improved bile acid metabolism. However, the regulation of SULT2A1 by LXR β through direct promoter binding is not well-supported by existing evidence, which suggests that LXR α may have a more dominant role in this process (PMID: 25028566). It would be beneficial to demonstrate whether CER(d18:1/18:1) preferentially binds LXR β over LXR α , possibly through computational modeling, and to assess the impact of ACER3 knockdown and LXR β knockdown on LXR α expression in both in vivo and in vitro models.
- 3. Relevance to Human Studies:** The study primarily uses female mice and HepG2 cells (derived from a male patient) to model the effects of ACER3 ablation and CER treatment. Given the observed sex differences in mice, the authors should address whether similar sex-specific effects might be expected in humans. Furthermore, the authors should explore differences in the expression levels of ACER3 and other ceramide-related genes between male and female humans to better understand potential sex-specific therapeutic implications. Additional analyses of SULT2A1 protein and mRNA expression levels in human samples, as used in Figure 1, would also help clarify these differences.
- 4. Experimental Controls and Data Interpretation:** Several experiments in the manuscript could benefit from additional control groups to clarify the observed effects. For instance, in Figure 2, including two groups of female Acer3 Δ Hep BDL mice treated with shCON and shSult2a1 would provide a clearer comparison of the impact of ACER3 ablation and SULT2A1 knockdown (Figure 2H–2T). The authors should also discuss potential species differences in Sult gene regulation, as Sult2a1 is known to be gender-specific in mice, with higher expression in female livers (Figure S3F).
- 5. Western Blot Analysis:** In Figures 3 and 5, the expression levels of internal controls in the Western blot analyses appear inconsistent across samples. The authors should confirm these findings by normalizing protein levels using an additional method, such as Coomassie Brilliant Blue (CBB) staining, to ensure the reliability of the data. Additionally, all Western blot data should be quantified.
- 6. Focus on B4GAT6:** While the study highlights ACER3, the authors should provide a rationale for focusing on this enzyme over B4GAT6, which also shows a strong correlation with serum cholestatic liver injury severity markers (SCSMs). Clarification on this point would enhance the interpretation of the data.
- 7. Regulation by Other Nuclear Receptors:** The manuscript mentions that Scd1 and Fasn are upregulated by CER(d18:1/18:1) treatment, but these genes are also regulated by other nuclear receptors, such as SREBP1 (Sterol

Regulatory Element-Binding Protein 1) and PPAR α . The authors should investigate whether these receptors, or their target genes, are also activated by CER(d18:1/18:1) treatment. This would provide a more comprehensive understanding of the pathway activation.

8. ACER3 Upregulation in Cholestasis: The authors should discuss the mechanisms underlying the observed upregulation of ACER3 in human cholestasis. Understanding why and how ACER3 is increased could provide important insights into the pathogenesis of cholestatic liver diseases and the potential therapeutic implications.

9. Discrepancy in CER(d18:1/18:1) Levels: The results indicate that CER(d18:1/18:1) treatment improves CLI, yet CER(d18:1/18:1) levels are elevated in CLI patients. The manuscript explains this discrepancy by suggesting that both the ceramide degrading and synthesizing systems are activated in CLI, with the synthesizing system prevailing. However, there is a lack of data to support this explanation. The authors should provide additional evidence to clarify this point. Furthermore, it would be valuable to explore whether there is a correlation between the severity of CLI in patients and the levels of CER(d18:1/18:1). Such data would strengthen the argument and provide a more comprehensive understanding of CER(d18:1/18:1)'s role in CLI.

Minor Comments:

1. Figure Legends and Data Presentation: The legends for Figures 3B and 3C lack sufficient detail and should be expanded to clarify their content and significance. Additionally, the data presented in Figures 4C and 4D are difficult to interpret. The authors should improve the clarity of these figures to better convey that lipidomics revealed a substantial decrease in lipid content, particularly triglycerols (TGs) and phospholipids (PLs), in the liver of Acer3^{fl/fl} female mice following BDL, and how this reduction was reversed in Acer3 Δ Hep female mice.

2. Expression Patterns of Sult2a Family: The expression patterns of the Sult2a gene family differ between the results obtained from transcriptomics analysis (Figure 2A) and those from qPCR (Figure S3A). The authors should address this discrepancy and provide an explanation for the differences observed between these two methodologies.

3. Typographical Errors: The manuscript contains several typographical errors that should be addressed before publication. For example, in lines 183-185, the sentence structure is unclear, and in lines 439-441, there are inconsistencies in the use of abbreviations.

4. Future Directions: The study raises interesting questions about the roles of other ceramides and ceramide enzymes in CLI. The authors might consider briefly discussing potential future studies that could explore these areas, particularly in the context of sex differences and the broader implications of ceramide metabolism in liver disease.

Reviewer #2

(Remarks to the Author)

• What are the noteworthy results?

The noteworthy results show a role for Acer3 and d18.1/18.1 ceramide in regulating bile acid sulfation via a direct interaction with LXRA α , with dramatic effects on severity of CLI in mouse models.

• Will the work be of significance to the field and related fields? How does it compare to the established literature? If the work is not original, please provide relevant references.

Overall, I found this manuscript to be well-written and the subject matter to be highly interesting and relevant to researchers in liver injury and sphingolipid signaling. The manuscript highlights an important role for Acer3 in cholestatic liver injury. These findings are highly novel. First, it is novel in that the role of Acer3 and other ceramidases in cell signaling and disease are not very well understood. Second, the general role of sphingolipids in cholestatic liver injury is also not very well appreciated.

• Does the work support the conclusions and claims, or is additional evidence needed?

The most consequential conclusions are very well-supported by the data. Specifically, the role of Acer3 and ceramides in mediating BA sulfation, the effect on expression of LXRA α and its targets, and the impact on the severity of BDL-induced disease.

I do not see any gaps that would prevent publication, however there were a few questions I felt were unanswered that may represent opportunities to improve the manuscript. Figure 1A and 1B reveal that several genes of the sphingolipid pathway are induced with CLI leading to increased levels of all but C24 and C26 ceramides. The correlation analysis in Fig. 1C points to ACER3 as a major factor. However, B4GALT6 is only slightly weaker as a candidate. Since upregulation of both genes have the same potential effect on ceramide levels, it would be good to see this gene followed up with in the BDL model. Especially since there are striking differences in the lipid profile of the BDL model when compared to the patient data, specifically only d18.1/18.1 cer changes in the BDL model versus nearly all ceramides in the patient data. Since ACER3 is induced in both patients and the BDL model, B4GALT6 may account for these differences.

• Are there any flaws in the data analysis, interpretation and conclusions? Do these prohibit publication or require revision?

Overall, the data was clear and well organized. One small complaint is the difficulty in evaluating the histology images as

they were presented. The resolution is too low in the pdf, and I was unable to locate higher resolution figure files. This only had an impact on the ability to evaluate data for Fig. panels 1H, 2D, 2L, 3D, 3N, 4E, 4G, 4I, and 5B. The oil red images in particular were not very informative, it was difficult to distinguish background staining from staining of lipid droplets.

- Is the methodology sound? Does the work meet the expected standards in your field?

The delivery of ceramide by IP with a carboxymethylcellulose vehicle is not typical. However, it was clearly effective for increasing hepatic d18.1/18.1 levels. Historically, it has been shown in the context of cell culture that ceramides with long acyl chains including d18.1/18.1 cer do not deliver well. Some studies, typically older ones, will employ a more soluble but artificial C6-ceramide. The C6-ceramide is rapidly remodeled leading to an increase in many sphingolipid species. In this historical context many researchers would avoid strategies that require exogenous delivery of ceramides. However, this strategy was clearly effective in this case. The point of these comments is that I feel that the approach would be more convincing to researchers in the sphingolipid field if the levels of all ceramide species were presented for this model instead of only d18.1/18.1 cer.

- Is there enough detail provided in the methods for the work to be reproduced?

The methods were sufficient and well-written.

Version 1:

Reviewer comments:

Reviewer #1

(Remarks to the Author)

Overall Review:

The authors have provided sincere and thorough responses to the reviewers' points, and the additional experiments are well-executed. In the previous review, we raised 9 Major Comments and 4 Minor Comments. Most of the Major Comments and all Minor Comments have been adequately addressed. However, I have additional comments regarding the authors' responses to some of the Major Comments. Addressing these issues would further enhance the quality of the manuscript.

Major Comments (Reviewer 1):

#1:

Response to Authors: Thank you for the detailed response. I have no further comments.

#2:

Response to Authors: Thank you for the detailed responses and the additional data.

- The authors state, "in ACER3-knockdown HepG2 cells, LXR α knockdown did not significantly reduce SULT2A1 expression" (Supplementary Figure 12I in the revised manuscript). However, the data in this figure appear to show a significant difference between shACER3-treated HepG2 cells treated with siCON and siLXR α . Could there be an error in the statistical analysis, or is there a specific rationale for interpreting these results as non-significant? Clarification on this point would be appreciated.

- Additionally, while the authors explain that LXR β is induced under ACER3 knockdown conditions, the underlying mechanism of this upregulation remains unclear. Could the authors elaborate on how ACER3 knockdown leads to increased LXR β expression? For example, is this mediated through transcriptional regulation, post-transcriptional stabilization, or other pathways? Please discuss.

#3:

Response to Authors: Thank you for the detailed response. The analysis of sex differences in each health condition (non-CLI and CLI) is thorough and well-presented. However, it would be beneficial to analyze differences within each sex according to health condition (e.g., "non-CLI vs. CLI in males" and "non-CLI vs. CLI in females") and to discuss the results in this context.

#4:

Response to Authors: Thank you for the detailed response. In the Western blot results (Figure C4-B), the expression level of Sult2a1 between the control and knockdown groups does not appear to differ significantly. It would be beneficial to quantify the Western blot results to provide clearer evidence supporting your conclusion.

#5:

Response to Authors: Thank you for the detailed response. I have no further comments.

#6:

Response to Authors: Thank you for the detailed response. I have no further comments.

#7:

Response to Authors: Thank you for the detailed response. I have no further comments.

#8:

Response to Authors: Thank you for the detailed response. I have no further comments.

#9:

Response to Authors: Thank you for the detailed response. I have no further comments.

Minor Comments:

Thank you for the detailed response. I have no further comments.

Reviewer #2

(Remarks to the Author)

• Does the work support the conclusions and claims, or is additional evidence needed? The most consequential conclusions are very well-supported by the data. Specifically, the role of Acer3 and ceramides in mediating BA sulfation, the effect on expression of LXRA and its targets, and the impact on the severity of BDL-induced disease. I do not see any gaps that would prevent publication, however there were a few questions I felt were unanswered that may represent opportunities to improve the manuscript.

Revised response: I feel that my initial comment was adequately addressed. I am satisfied with the revised Results and Discussion section.

• Are there any flaws in the data analysis, interpretation and conclusions? Do these prohibit publication or require revision? Overall, the data was clear and well organized. One small complaint is the difficulty in evaluating the histology images as they were presented. The resolution is too low in the pdf, and I was unable to locate higher resolution figure files. This only had an impact on the ability to evaluate data for Fig. panels 1H, 2D, 2L, 3D, 3N, 4E, 4G, 4I, and 5B. The oil red images in particular were not very informative, it was difficult to distinguish background staining from staining of lipid droplets.

Revised response: I am satisfied my initial comment was adequately addressed with the availability of higher quality images and the revised figure legends.

• Is the methodology sound? Does the work meet the expected standards in your field? The delivery of ceramide by IP with a carboxymethylcellulose vehicle is not typical. However, it was clearly effective for increasing hepatic d18.1/18.1 levels. Historically, it has been shown in the context of cell culture that ceramides with long acyl chains including d18.1/18.1 cer do not deliver well. Some studies, typically older ones, will employ a more soluble but artificial C6-ceramide. The C6-ceramide is rapidly remodeled leading to an increase in many sphingolipid species. In this historical context many researchers would avoid strategies that require exogenous delivery of ceramides. However, this strategy was clearly effective in this case. The point of these comments is that I feel that the approach would be more convincing to researchers in the sphingolipid field if the levels of all ceramide species were presented for this model instead of only d18.1/18.1 cer.

Revised response: I appreciate the inclusion of more detailed lipidomics data, and I am sure these will be appreciated by the readers as well. I am more than satisfied that the newly incorporated data addresses my initial comments.

Version 2:

Reviewer comments:

Reviewer #1

(Remarks to the Author)

Thank you for the detailed response. I have no further comments.

Dear Reviewers,

The following are our point-by-point responses to the requests, questions, concerns, and comments. The corresponding modifications in our revised manuscript are highlighted in red-colored text. For each response in this section, the data titled Figure C, which supports our explanations has been provided in this rebuttal letter. Corresponding data have also been incorporated into the revised manuscript.

Reviewer #1 (Remarks to the Author):

The manuscript titled “Targeting ACER3 Attenuates Cholestasis by Mitigating Bile Acid Overload via Unsaturated Ceramide-mediated LXR β Signaling Transduction” significantly contributes to our understanding of the role of ceramide metabolism in cholestatic liver injury (CLI). The study explores the pathological role of ACER3 and its substrate CER(d18:1/18:1) in CLI, revealing that the ablation of ACER3 in female mice ameliorates CLI through the LXR β -SULT2A1 axis. The findings suggest potential therapeutic targets for CLI, particularly through the modulation of ceramide metabolism. However, several aspects of the study require further attention and clarification.

Major Comments:

1. Sex Differences in ACER3 Function: The study identifies a significant sex difference in the effects of ACER3 ablation, with protective effects observed only in female mice. This difference is attributed to the higher basal expression of Sult2a1 in females. However, the manuscript does not fully explore the implications of this sex difference for potential human therapies. The authors should provide additional data and discussion on SULT2A1 expression levels in human liver samples, particularly regarding potential sex differences. This would strengthen the relevance of the findings for clinical applications.

Response: We thank the reviewer for this insightful suggestion. Previous studies have shown a notable sex difference in *Sult2a1* expression in mouse liver, with females exhibiting higher expression than males (PMID: 16807285). However, the sex differences in *SULT2A1* expression in the human liver remain unclear. To address this, we first conducted an *in silico* analysis using publicly available datasets from the GTEx Portal (<https://www.gtexportal.org/>). Our analysis revealed that *SULT2A1* mRNA expression did not significantly differ between female and male humans under normal conditions (Figure C1A). To validate these findings, we measured *SULT2A1* mRNA and protein levels in the collected liver tissues from patients with and without cholestatic liver injury (CLI). Consistent with the *in silico* results, there was no significant difference in *SULT2A1* expression between male and female human liver tissues in the non-CLI group. In the CLI liver tissues, *SULT2A1* expression was similarly reduced in both sexes (Figures C1B-C1G). These findings suggest that *SULT2A1* expression is comparable between men and women under normal conditions, and cholestasis impairs *SULT2A1* expression equally in both sexes (Figures C1B, C1F, and C1G). Our findings align with previous observations that the human liver primarily expresses *SULT2A1* (PMID: 7678732) and that sulfation activities in the human liver appear similar between females and males (PMID: 25738837). Therefore, the sex difference in *Sult2a1* expression observed in mouse liver does not extend to the human liver. This strengthens the potential clinical relevance of our findings, indicating that targeting *SULT2A1* pathways may benefit both men and women with cholestasis. The detailed data of these new findings have been added as Figure 7, and the Results and Discussion Sections have been revised accordingly.

Figure C1

2. Mechanistic Insights and Molecular Interactions: The manuscript discusses CER(d18:1/18:1) as an agonist of LXR β , leading to the upregulation of SULT2A1 and improved bile acid metabolism. However, the regulation of SULT2A1 by LXR β through direct promoter binding is not well-supported by existing evidence, which suggests that LXR α may have a more dominant role in this process (PMID: 25028566). It would be beneficial to demonstrate whether CER(d18:1/18:1) preferentially binds LXR β over LXR α , possibly through computational modeling, and to assess the impact of ACER3 knockdown and LXR β knockdown on LXR α expression in both in vivo and in vitro models.

Response: We thank the reviewer for this valuable suggestion regarding the mechanistic insights. Both LXR α and LXR β have been shown to bind the promoter of the SULT2A gene, acting as transcriptional regulators (PMID: 17256725). In agreement with this, our study demonstrated that LXR β plays a role in controlling SULT2A1 expression in the cholestatic liver (Figures 3I-3M in our revised manuscript). While existing evidence (PMID: 25028566) suggests that LXR α may play a more dominant role in regulating *SULT2A1* in vitro, the distinct regulatory roles of LXR α and LXR β on *SULT2A1* expression in vivo, especially under disease conditions like cholestasis, still require further exploration. Our findings highlight the role of CER(d18:1/18:1) in

the regulation of *SULT2A1* expression by interacting with LXR β . To further explore whether CER(d18:1/18:1) also interacts with LXR α to regulate *SULT2A1* expression, as suggested by the reviewer, we first conducted computational modeling to compare the binding affinity of CER(d18:1/18:1) to LXR α and LXR β . Our modeling results revealed that CER(d18:1/18:1) had a higher binding affinity for LXR β than for LXR α across all available protein structures, indicating a preference for LXR β (Figures C2A and C2B). To further clarify the roles of LXR α and LXR β in mediating the effects of ACER3 and CER(d18:1/18:1) on *SULT2A1* expression, we examined the impact of *ACER3/Acer3* knockdown and *LXR β /Lxr β* knockdown on *LXR α /Lxr α* expression in both *in vitro* and *in vivo* models. We found that double knockdown of *ACER3/Acer3* and *LXR β /Lxr β* had no significant effect on *LXR α /Lxr α* expression in either HepG2 cells or female mouse livers (Figures C2D and C2E). Furthermore, in *ACER3*-knockdown HepG2 cells, *LXR α* knockdown did not significantly reduce *SULT2A1* expression compared to *LXR β* knockdown (Figures C2F-C2I). These findings suggest that LXR β plays a more specific and pivotal role in mediating the effects of ACER3 and CER(d18:1/18:1) compared to LXR α . The detailed data of these new findings have been added to Supplementary Figures 8 and 12, as well as Tables S3 and S4, and the Results and Discussion Sections have been revised accordingly.

Figure C2**A** Grid Scores between human LXR β and CER(d18:1/18:1) (kcal/mol)

PDB ID	pose	Grid Score	Grid vdW	Grid es	Internal energy
5i4v	1	-117.30	-108.83	-8.48	19.35
6s4t	1	-110.54	-103.86	-6.67	17.23
1pq6-A	1	-105.76	-103.69	-2.07	18.87
	2	-102.77	-88.36	-14.41	20.34
4dk7	1	-92.90	-91.33	-1.57	27.18
4dk8	1	-81.48	-77.88	-3.60	17.17
5kya	1	-77.42	-74.63	-2.79	26.39
	2	-70.93	-69.57	-1.36	44.09
5hjp	1	-77.03	-73.49	-3.54	25.21
	2	-71.75	-70.59	-1.16	25.11
4nqa	1	-74.40	-69.25	-5.15	40.13
	2	-72.93	-69.61	-3.32	46.90
5kyj	1	-73.86	-67.66	-6.20	19.88
	2	-72.20	-66.87	-5.33	23.70
6k9m	1	-70.86	-66.15	-4.71	26.79
	2	-69.62	-64.31	-5.31	21.12
	3	-69.34	-64.14	-5.20	19.76
6jio-A	1	69.22	68.17	1.05	64.07
6s4n	1	-63.69	-63.85	0.17	29.73
	2	-53.49	-53.69	0.19	36.07
	3	-50.54	-48.94	-1.60	52.99
6s4u	1	-55.26	-52.35	-2.91	53.74
6s5k	1	-54.84	-52.50	-2.33	45.22
3l0e	1	-54.74	-50.06	-4.68	20.85
	2	-53.90	-49.22	-4.68	19.52
	3	-53.64	-48.96	-4.68	17.56
	4	-53.57	-48.89	-4.68	18.74
	5	-53.01	-48.33	-4.68	22.28
5jy3	1	52.50	54.05	-1.54	97.68
1p8d	1	-57.03	-54.06	-2.97	57.21
	2	-51.34	-50.30	-1.04	54.05
	3	-48.98	-45.79	-3.19	52.52
	4	-44.35	-42.38	-1.97	26.31
1upv	1	-59.90	-57.73	-2.17	32.92
	2	-39.69	-37.65	-2.04	60.10
1pq9	1	51.79	51.92	-0.13	76.81
1upw	1	-48.25	-41.86	-6.40	56.75
6k9h	1	-35.15	-40.54	5.40	50.34
1pqc-B	1	-25.96	-25.68	-0.27	22.47
4rak	1	-16.98	-15.89	-1.09	27.52
3kfc	1	-6.26	-6.03	-0.23	74.18
	2	-5.26	0.69	-5.95	31.07
6k9g	1	35.45	38.61	-3.16	57.48

Figure C2

3. Relevance to Human Studies: The study primarily uses female mice and HepG2 cells (derived from a male patient) to model the effects of ACER3 ablation and CER treatment. Given the observed sex differences in mice, the authors should address whether similar sex-specific effects might be expected in humans.

Furthermore, the authors should explore differences in the expression levels of ACER3 and other ceramide-related genes between male and female humans to better understand potential sex-specific therapeutic implications. Additional analyses of SULT2A1 protein and mRNA expression levels in human samples, as used in Figure 1, would also help clarify these differences.

Response: We appreciate the reviewer for raising this important point regarding the potential sex-specific effects of *ACER3* ablation and CER(d18:1/18:1) treatment in human cholestasis. To explore this further, we conducted an *in silico* analysis of CER-related gene expression in the liver tissues of healthy male and female humans using the GTEx dataset (<https://www.gtexportal.org/>), as well as the clinical samples we collected for this study. Our findings revealed no significant differences in the mRNA expression of CER-related genes between female and male livers under either non-cholestasis or cholestasis conditions (Figures C3A and C3B). Additionally, CER measurements in the collected clinical samples showed that most CER(d18:1) species did not differ significantly between sexes under normal conditions (Figure C3C). Although cholestasis decreased CER(d18:1/26:0) and CER(d18:1/26:1) in female livers compared to male livers (Figure C3D), cholestasis-induced upregulation of *ACER3* and CER(d18:1/18:1) was comparable between female and male patients (Figures C3E and C3F). Moreover, as we discussed in our response to your Comment 1, our examination of *SULT2A1* expression using the same datasets and clinical samples revealed that the expression levels of *SULT2A1* were similar between men and women, with cholestasis impairing *SULT2A1* expression similarly in both sexes (Figure C1). Together, these results suggest that CER-related gene expression and the regulation of *ACER3* and CER(d18:1/18:1) are not sex-specific in humans, either under non-cholestasis or cholestasis conditions. Therefore, it is unlikely to expect that the regulatory roles of *ACER3* and CER(d18:1/18:1) in cholestasis differ between male and female humans. Moreover, *ACER3* knockdown effectively reduced BA toxicity by upregulating *SULT2A1* in HepG2 cells, which were originally derived from a male patient (PMID: 24160292). This supports that targeting *ACER3* to increase CER(d18:1/18:1)

may be an effective therapeutic strategy for attenuating cholestatic liver injury in both male and female patients. The detailed data of these new findings have been added as Figure 7, and the Results and Discussion sections have been revised accordingly.

Figure C3

4. Experimental Controls and Data Interpretation: Several experiments in the manuscript could benefit from additional control groups to clarify the observed effects. For instance, in Figure 2, including two groups of female Acer3ΔHep BDL mice treated with shCON and shSult2a1 would provide a clearer comparison of the impact of ACER3 ablation and SULT2A1 knockdown (Figure 2H–2T). The authors should also discuss potential species differences in Sult gene regulation, as Sult2a1 is known to be sex-specific in mice, with higher expression in female livers (Figure S3F).

Response: We appreciate the reviewer's insightful comments and suggestions regarding additional control groups and species differences in Sult gene regulation. To address the reviewer's suggestion for additional control groups, we have included the data for *Acer3*^{fl/fl} female mice treated with either shCON or shSult2a1 vectors via adeno-associated viruses in Figure 2. Knockdown of *Sult2a1* in *Acer3*^{fl/fl} female mice was validated at both mRNA and protein levels (Figures C4A-C4C). The knockdown of *Sult2a1* significantly reduced bile acid sulfates (BA-sulfates) and led to the accumulation of BAs, thereby exacerbating CLI, as evidenced by worsened hepatic necrosis, inflammation, and fibrosis (Figures C4D-C4T). These results underscore the protective role of Sult2a1-catalyzed BA sulfation in preventing CLI in female mice. *Acer3*^{ΔHep} female mice exhibited enhanced Sult2a1-catalyzed BA sulfation and attenuated CLI compared to *Acer3*^{fl/fl} female mice (Figures C4D-C4J). Importantly, the knockdown of *Sult2a1* in *Acer3*^{ΔHep} female mice abolished the enhanced BA sulfation and led to a similar degree of CLI as observed in *Sult2a1*-knockdown *Acer3*^{fl/fl} female mice. This demonstrates that loss of *Sult2a1* abrogates the protective effects of *Acer3* ablation in female mice, therefore, these protective effects are mediated through Sult2a1-catalyzed BA sulfation (Figure C4). These new findings have been added to Figure 2 and Supplementary Figure 6, and the Results and Discussion sections have been updated accordingly.

Regarding species differences in Sult gene regulation, it is well-established that the expression of the Sult2a family is sex-specific in various rodent models, with higher expression in female livers of FVB mice, C57BL/6 mice, and Fischer F-344 rats (PMID: 28661152, PMID: 10850433, PMID: 18725207). However, this sex-specific expression is absent in guinea pigs and hamsters (PMID: 2783155), indicating species-specific variation in Sult regulation. In humans, the liver expresses only *SULT2A1* (PMID: 7678732) and sulfation activities in the human liver have been found to be similar between females and males (PMID: 25738837). Our findings are consistent with these previous studies, as we observed sex-specific expression of the *Sult2a1* gene in

C57BL/6 mice but not in human liver samples. These observations further support the notion that while Sult2a1 plays a sex-specific role in rodents, this does not extend to humans. We have incorporated these points into the Discussion section to highlight the importance of considering species-specific differences in Sult gene regulation.

Figure C4

Figure C4

5. Western Blot Analysis: In Figures 3 and 5, the expression levels of internal controls in the Western blot analyses appear inconsistent across samples. The authors should confirm these findings by normalizing protein levels using an additional method, such as Coomassie Brilliant Blue (CBB) staining, to ensure the reliability of the data. Additionally, all Western blot data should be quantified.

Response: We appreciate the reviewer's suggestions regarding the Western blot analysis. In response, we have taken several steps to ensure the reliability and accuracy of the data presented in Figures 3 and 5. For Figures 3B, 3C, 3F, 3I, and 3V, we repeated the Western blot analysis for Coomassie Brilliant Blue (CBB) staining using the same protein samples and identical loading amounts as in the original experiments. For Figure 3H, due to the depletion of frozen tissue previously used for cytoplasmic and nuclear fraction separation, we repeated the animal experiments to extract nuclear and cytoplasmic components and performed the Western blot accordingly. Similarly, for Figures 5J and 5N, we had used up the initial protein samples, so we performed Western blot analysis using newly extracted protein from the same tissues to replicate the previous findings. To further validate the results, we performed CBB staining to demonstrate the uniformity of protein loading across samples (Figure C5). Additionally, all Western blot data were quantified by normalizing to the internal control, and the quantification results were provided below each corresponding band. Results of the statistical analysis were provided in the Source Data Files. The updated results are consistent with those from our previously submitted manuscript, affirming the reliability of the data. The new Western blot data, with accompanying CBB staining, have replaced the original blots and have been included in Supplementary Figure 14.

Figure C5

6. Focus on B4GAT6: While the study highlights ACER3, the authors should provide a rationale for focusing on this enzyme over B4GAT6, which also shows a strong correlation with serum cholestatic liver injury severity markers (SCSMs). Clarification on this point would enhance the interpretation of the data.

Response: We appreciate the reviewer’s suggestion regarding the focus on ACER3 over B4GAT6. While both B4GAT6 and ACER3 were upregulated in cholestasis and showed correlations with serum cholestatic liver injury severity markers (SCSMs) in our study (Figure 1 in our revised manuscript), the correlation between ACER3

expression and SCSMs was stronger and more consistent across a broader range of markers compared to *B4GALT6* (Figure S1B in our revised manuscript). Specifically, in our newly added analyses of CER-metabolizing enzymes in patients with CLI, *ACER3* exhibited correlations with a greater number of SCSMs than *B4GALT6* (Figure 1C in our revised manuscript), suggesting that *ACER3* may play a more central role in the pathophysiology of CLI. This consistent upregulation of *ACER3* further reinforces its significance in cholestasis and the rationale for our study to focus on *ACER3*. *B4GALT6* plays a role in the synthesis of lactosylceramides (LacCERs) and has been implicated in inflammatory processes (PMID: 36982367 and PMID: 25216636). We found that *B4galt6* was upregulated in the liver of humans and mice with cholestasis (Figures C6A-C6D). To further investigate the function of *B4galt6* and in comparison to *Acer3*, we conducted functional experiments using female mice. We found that LacCERs and glucosylceramides (GluCERs) were elevated in female mice with cholestasis (Figure C6E). Treatment of *D-threo*-1-phenyl-2-decanoylamino-3-morpholino-1-propanol (*D*-PDMP), the pharmacological inhibitor of *B4galt6* (PMID: 25216636), reduced inflammation and fibrosis without affecting *Sult2a1* expression (Figures C6F-C6Q). These findings suggest that *B4GALT6* may modulate cholestasis through different mechanisms, perhaps more related to inflammation, while *ACER3* has a crucial regulatory role in BA metabolism and CER signaling pathways. Thus, while *B4GALT6* is an intriguing enzyme for future investigation, the centrality of *ACER3* in regulating BA metabolism and its stronger correlation with SCSMs justify the focus of this study on *ACER3*. We encourage further exploration of *B4GALT6* in the context of cholestasis, particularly concerning its specific molecular mechanisms. The data of these new findings of *B4galt6* functions have been added to Supplementary Data 13 to inspire and facilitate future exploration. The Results and Discussion sections have also been updated to clarify these points and reinforce the focus on *ACER3* in this study.

Figure C6

7. Regulation by Other Nuclear Receptors: The manuscript mentions that *Scd1* and *Fasn* are upregulated by CER(d18:1/18:1) treatment, but these genes are also regulated by other nuclear receptors, such as SREBP1 (Sterol Regulatory Element-Binding Protein 1) and PPARα. The authors should investigate whether these receptors, or their target genes, are also activated by CER(d18:1/18:1)

treatment. This would provide a more comprehensive understanding of the pathway activation.

Response: We appreciate the reviewer's valuable suggestion to further explore the molecular mechanisms underlying the upregulation of *Scd1* and *Fasn* by CER(d18:1/18:1) treatment. In response, we investigated whether other nuclear receptors, including *SREBP1* and *PPAR α* , could also be involved in this regulation. Our results showed that the mRNA and protein levels of *Srebp1* and *Ppara* were not significantly affected by CER(d18:1/18:1) treatment, indicating that these nuclear receptors are unlikely to mediate the observed upregulation of *Scd1* and *Fasn* (Figures C7A and C7B). This suggests that the effects of CER(d18:1/18:1) treatment on these lipogenic genes are not driven by *Srebp1* or *Ppara*. On the other hand, Lxr β has been shown to regulate *Scd1* and *Fasn* expression (PMID: 29904174). In our study, we found that *Lxr β* knockdown in *Acer3* ^{Δ Hep} female mice with CLI abolished the upregulation of *Scd1* and *Fasn*, reinforcing the role of *Lxr β* in mediating these effects (Figure S7C in our revised manuscript). To further confirm the role of *Lxr β* in CER(d18:1/18:1)-mediated effects, we examined whether *Lxr β* knockdown could also negate the upregulation of *Scd1* and *Fasn* in CER(d18:1/18:1)-treated mice with CLI. Our results showed that *Lxr β* knockdown not only diminished hepatic lipid content but also abolished the upregulation of *Scd1* and *Fasn* induced by CER(d18:1/18:1) treatment (Figures C7C-C7E). These findings confirm that Lxr β , rather than *Srebp1* or *Ppara*, is essential for the CER(d18:1/18:1)-induced upregulation of *Scd1* and *Fasn*, and plays a critical role in maintaining hepatic lipid homeostasis under CLI. We have added these new data to Figure 6 and Supplementary Figure 8 and revised the Results and Discussion Sections accordingly.

Figure C7

8. ACER3 Upregulation in Cholestasis : The authors should discuss the mechanisms underlying the observed upregulation of ACER3 in human cholestasis. Understanding why and how ACER3 is increased could provide important insights into the pathogenesis of cholestatic liver diseases and the potential therapeutic implications.

Response: We appreciate the reviewer's insightful suggestion regarding the mechanisms underlying *ACER3* upregulation in human cholestasis. To investigate this, we conducted online data mining using publicly available datasets, including hTFtarget, ChIP_Atlas, GTRD, ENCODE, and FIMO_JASPAR. Our analysis identified several potential transcriptional factors that may regulate *ACER3* expression, including CTCF, ELF1, GABPA, PAX5, SP1, STAT1, TCF12, ETS1, EGR1, ZBTB7A, ZNF263, STAT3, and SP4 (Figure C8). Among these candidates, SP1, EGR1, and STAT3 are particularly notable and could play key roles in increasing *ACER3* expression, as they were reported to be upregulated in response to cholestasis (PMID: 30063921, PMID: 33746084, and PMID: 25450715). Interestingly, these factors are also known to regulate CER

metabolism. SP1 has been shown to upregulate glucosylceramide synthase (GCS) and neutral sphingomyelinase-2 (SMPD3) in cancer cells (PMID: 15342415 and PMID: 26512957), EGR1 is linked to the regulation of UDP glucose ceramide glucosyltransferase (UGCG) and sphingosine kinase 2 (SPHK2) (PMID: 34843781 and PMID: 28939554), and STAT3 is capable of upregulating N-acylsphingosine amidohydrolase 2 (ASAH2) (PMID: 36630483). These suggest that cholestasis may drive *ACER3* upregulation through multiple regulatory pathways involving these transcription factors. These findings provide a plausible explanation for the upregulation of *ACER3* in cholestasis and reinforce the role of *ACER3* in CER metabolism under cholestasis. We have included these new data in Supplementary Figure 1 and revised the Results and Discussion Sections to reflect these insights.

Figure C8

9. Discrepancy in CER(d18:1/18:1) Levels: The results indicate that CER(d18:1/18:1) treatment improves CLI, yet CER(d18:1/18:1) levels are elevated in CLI patients. The manuscript explains this discrepancy by suggesting that both the ceramide degrading and synthesizing systems are activated in CLI, with the synthesizing system prevailing. However, there is a lack of data to support this explanation. The authors should provide additional evidence to clarify this point. Furthermore, it would be valuable to explore whether there is a correlation

between the severity of CLI in patients and the levels of CER(d18:1/18:1). Such data would strengthen the argument and provide a more comprehensive understanding of CER(d18:1/18:1)'s role in CLI.

Response: We appreciate the reviewer's insightful suggestions regarding the discrepancy in CER(d18:1/18:1) levels and their role in CLI. In our original submission, we proposed that both CER-degrading and CER-synthesizing systems are activated in CLI, with the synthesizing system prevailing. To further clarify this explanation, we measured the protein levels of CER-related enzymes and the levels of CER degradation products, sphingosine (SPH) and sphingosine-1-phosphate (S1P), during the revision process. We found that enzymes contributing to CER generation were upregulated by cholestasis, including DEGS2 in humans and *Smpd3* and *Cers3* in mice (Figures C9A-C9E), indicating activation of CER generation by cholestasis. Concurrently, the upregulation of *ACER3/Acer3* and *ASAHI* suggested activation of CER degradation (Figure 1B in our revised manuscript). However, while overall CER levels were increased in response to cholestasis, the levels of SPH, a CER degradation product, were significantly decreased (Figure C9F). This suggests that while CER generation was upregulated, the activation of degradation catalyzed by *ACER3/Acer3* and *ASAHI* might have been insufficient to significantly affect SPH levels, or SPH may have been utilized as a substrate for the salvage synthesis of CERs (PMID: 29165427), leading to decreased SPH levels. Thus, CER production likely predominated over degradation in the cholestatic liver. Consequently, although *Acer3* was upregulated to hydrolyze CER(d18:1/18:1), the overwhelming production of CER(d18:1/18:1) exceeded its hydrolytic capacity, resulting in net accumulation of CER(d18:1/18:1) in the cholestatic liver. Our observation that *Acer3* ablation further increased CER(d18:1/18:1) in the cholestatic liver (Figure 4A in our revised manuscript) underscores this dynamic, indicating that *ACER3* upregulation acts to limit the buildup of CER(d18:1/18:1) in the cholestatic liver.

To strengthen the clinical significance of CER(d18:1/18:1) in CLI, we performed

correlation analyses between hepatic CER(d18:1/18:1) levels and SCSMs in patients with CLI. Interestingly, we found that hepatic CER(d18:1/18:1) levels were negatively correlated with SCSMs, including ALT, AST, TBIL, and DBIL (Figure C9G). These findings suggest that lower hepatic CER(d18:1/18:1) levels are associated with more severe CLI, thereby reinforcing the protective role of CER(d18:1/18:1) in mitigating CLI severity. We believe that these new data provide substantial support for our initial hypothesis and clarify the complex dynamics of CER metabolism in CLI.

We have added these findings to Figure 1 and Supplementary Figures 1 and 2, and have revised the Results and Discussion Sections accordingly.

Figure C9

Minor Comments:

1. Figure Legends and Data Presentation: The legends for Figures 3B and 3C lack sufficient detail and should be expanded to clarify their content and significance. Additionally, the data presented in Figures 4C and 4D are difficult to interpret. The authors should improve the clarity of these figures to better convey that lipidomics revealed a substantial decrease in lipid content, particularly triglycerols (TGs) and phospholipids (PLs), in the liver of *Acer3^{fl/fl}* female mice following BDL, and how this reduction was reversed in *Acer3^{ΔHep}* female mice.

Response: We appreciate the reviewer for highlighting these points related to data presentation. **Figure 3B** in our revised manuscript presents the results of the Western blot for LXRβ in the liver of *Acer3^{fl/fl}* and *Acer3^{ΔHep}* female mice under sham conditions. The data show that the protein levels of LXRβ were significantly upregulated in the liver of *Acer3^{ΔHep}* compared to *Acer3^{fl/fl}* female mice under sham conditions ($P = 0.0183$). **Figure 3C** in our revised manuscript shows the Western blot results for LXRβ in the liver of *Acer3^{fl/fl}* and *Acer3^{ΔHep}* female mice under both sham and BDL conditions. The protein levels of LXRβ were significantly higher in *Acer3^{ΔHep}* female mice compared to *Acer3^{fl/fl}* female mice under both conditions ($P = 0.0233$ for sham, $P = 0.0007$ for BDL). We have revised the annotation of these figures to clarify these results.

For **Figures 4C** and **4D** in our revised manuscript, we acknowledge that the data interpretation could be clearer. These figures present the scoring and loading plots of the principal component analysis (PCA) of the lipidomic data. **Figure 4C** illustrates the discrimination in lipid content in the liver of *Acer3^{fl/fl}* and *Acer3^{ΔHep}* female mice under both sham and BDL conditions. To enhance the clarity of **Figure 4D**, we have added arrow labels indicating the enriched lipid content in the indicated mouse groups. Furthermore, we have supplemented the PCA plots with volcano plots (**Figures 4E-4H** in our revised manuscript) to better highlight differences in hepatic lipid species between the groups, with specific emphasis on triglycerides (TGs) and phospholipids (PLs). The volcano plots illustrate the following: **Figure 4E**: TGs (pink dots) and PLs

(blue dots) were substantially reduced in the liver of *Acer3^{fl/fl}* female mice following BDL compared to sham surgery. **Figure 4F:** *Acer3* ablation did not significantly alter the hepatic lipidome under sham conditions. **Figure 4G:** TGs were substantially reduced, while some PLs increased in the liver of *Acer3^{ΔHep}* female mice following BDL. **Figure 4H:** Certain PLs were significantly higher in *Acer3^{ΔHep}* female mice compared to *Acer3^{fl/fl}* female mice after BDL. These additional data illustrate that cholestasis notably reduced hepatic lipid content, particularly TGs and PLs, and *Acer3* ablation helped reverse the reduction of PLs in cholestatic livers. We believe that these modifications improve the clarity of the results and support the conclusions drawn. The legends for Figure 4 and the corresponding text in the Results Section have been updated accordingly.

2. Expression Patterns of Sult2a Family: The expression patterns of the Sult2a gene family differ between the results obtained from transcriptomics analysis (Figure 2A) and those from qPCR (Figure S3A). The authors should address this discrepancy and provide an explanation for the differences observed between these two methodologies.

Response: Thank you for this insightful comment. During the revision, we identified

an error in Figure 2A in our previous manuscript, where the Y-axis values were incorrectly labeled. We have since corrected this mistake. Despite the correction, the *Sult2a* genes remain significantly upregulated in the liver of *Acer3*^{ΔHep} female mice. We apologize for this oversight and have uploaded the original data and deposited RNA sequencing raw data to the Sequence Read Archive database for transparency. Regarding the observed discrepancy between the transcriptomics analysis (RNA sequencing) and qPCR results of *Sult2a* gene expression (Figures 2A and S4A in our revised manuscript), we recognize that slight variations can arise between these two methodologies probably due to inherent differences in their sensitivity and technical approaches. RNA sequencing is a high-throughput method that provides a global view of gene expression but can be influenced by factors such as sequencing depth, normalization techniques, and the detection of low-abundance transcripts (PMID: 31341269). In contrast, qPCR is a more targeted, sensitive, and specific method, particularly when measuring individual genes like those in the *Sult2a* family. Another contributing factor is that RNA sequencing and qPCR can detect different transcripts of the same gene, potentially leading to variations in their expression profiles (PMID: 33276803). Furthermore, the sample sets used for each method differed: we used 4 pairs of mouse samples for RNA sequencing and another 8 pairs for qPCR validation. Differences in sample size and variability between individual samples may also account for some of the observed discrepancies. Nevertheless, both methods consistently identified that *Sult2a1* was upregulated in the liver of *Acer3*^{ΔHep} female mice. This upregulation was further validated at the protein level by Western blot, reinforcing the reliability of our findings. We hope this explanation addresses your concerns, and we thank you for highlighting this important point.

3. Typographical Errors: The manuscript contains several typographical errors that should be addressed before publication. For example, in lines 183-185, the sentence structure is unclear, and in lines 439-441, there are inconsistencies in the use of abbreviations.

Response: Thank you for pointing out these typographical issues. We have carefully reviewed the manuscript and corrected the following:

Lines 183-185: The unclear sentence structure has been rewritten due to adding new data.

Lines 439-441: Inconsistencies in the use of abbreviations have been addressed. Specifically, abbreviations are now consistently used throughout the manuscript.

Additionally, we performed a thorough proofreading of the manuscript to correct any other typographical errors, ensuring consistent and professional language throughout the paper. The corrections have been highlighted in red for ease of reference.

4. Future Directions: The study raises interesting questions about the roles of other ceramides and ceramide enzymes in CLI. The authors might consider briefly discussing potential future studies that could explore these areas, particularly in the context of sex differences and the broader implications of ceramide metabolism in liver disease.

Response: Thank you for this suggestion. We agree that further exploration into the roles of other CERs and CER-metabolizing enzymes in CLI could provide more valuable insights. In response, we have modified the Discussion Section by briefly outlining potential future directions:

While our study highlights the specific role of ACER3-catalyzed CER(d18:1/18:1) hydrolysis in mitigating bile acid overload in CLI, the broader landscape of CER metabolism in CLI remains underexplored. In particular, special attention may be given to enzymes such as ASAH1, B4GALT6, and DEGS2, along with their substrates, as these were found to be dysregulated by CLI (Figure 1 in our revised manuscript). For instance, B4GALT6 is involved in the synthesis of LacCERs and has been implicated in inflammatory processes (PMID: 36982367, PMID: 25216636). Our ongoing study

found that B4galt6, LacCERs, and GluCERs were elevated in mice with cholestasis, and the pharmacological inhibition of B4galt6, using D-PDMP (PMID: 25216636), reduced inflammation and fibrosis without affecting *Sult2a1* expression in female mice (Figure C6). These findings suggest that CER metabolism is tightly regulated by cholestasis, with enzymes such as ACER3, B4GALT6, and others playing critical roles in influencing CLI through distinct mechanisms. Furthermore, while our findings suggest that the sex-specific response to *Acer3* ablation in mice is not expected in humans, large-scale studies are still needed to further confirm the sex-specific differences in CER metabolism. Future studies should pay particular attention to sex differences, examining whether the sex-specific effects observed in animal models also apply to humans. Lastly, building on the previous study demonstrated BA-regulated signaling influences CER metabolism (PMID: 33938457), our study reveals the role of CER in regulating BA metabolism, underscoring the importance of exploring the regulatory interactions between CER and other metabolic pathways to uncover broader pathophysiological roles of metabolic cross-regulation in liver diseases.

We hope these additions highlight the broader potential of CER research and encourage exploration of these critical pathways.

Reviewer #2 (Remarks to the Author):

• What are the noteworthy results?

The noteworthy results show a role for *Acer3* and d18:1/18:1 ceramide in regulating bile acid sulfation via a direct interaction with LXRalpha, with dramatic effects on severity of CLI in mouse models.

Response: Thank you for the positive feedback on our study. We are pleased that you found the role of ACER3 and ceramide(d18:1/18:1) in regulating bile acid sulfation through interaction with LXR β to be noteworthy, particularly in its dramatic effects on the severity of CLI in our mouse models. This feedback reinforces the significance of our findings in uncovering the molecular mechanisms underlying BA metabolism and

its potential implications for future therapeutic strategies.

- **Will the work be of significance to the field and related fields? How does it compare to the established literature? If the work is not original, please provide relevant references.**

Overall, I found this manuscript to well-written and the subject matter to be highly interesting and relevant to researchers in liver injury and sphingolipid signaling. The manuscript highlights an important role for Acer3 in cholestatic liver injury. These findings are highly novel. First, it is novel in that the role of Acer3 and other ceramidases in cell signaling and disease are not very well understood. Second, the general role of sphingolipids in cholestatic liver injury is also not very well appreciated.

Response: Thank you for your positive and encouraging feedback. We are delighted that you found our manuscript to be well-written and that the topic is both highly relevant and novel within the fields of liver injury and sphingolipid signaling. The recognition of the significance of ACER3's role in CLI, as well as the broader implications for ceramidase-mediated cell signaling and sphingolipid involvement in liver diseases, strengthens our belief in the contribution this work makes to advancing the understanding of these mechanisms.

- **Does the work support the conclusions and claims, or is additional evidence needed?**

The most consequential conclusions are very well-supported by the data. Specifically, the role of Acer3 and ceramides in mediating BA sulfation, the effect on expression of LXRalpha and its targets, and the impact on the severity of BDL-induced disease. I do not see any gaps that would prevent publication, however there were a few questions I felt were unanswered that may represent opportunities to improve the manuscript. Figure 1A and 1B reveal that several genes of the sphingolipid pathway are induced with CLI leading to increased levels

of all but C24 and C26 ceramides. The correlation analysis in Fig. 1C points to ACER3 as a major factor. However, B4GALT6 is only slightly weaker as a candidate. Since upregulation of both genes have the same potential effect on ceramide levels, it would be good to see this gene followed up with in the BDL model. Especially since there are striking differences in the lipid profile of the BDL model when compared to the patient data, specifically only d18.1/18.1 cer changes in the BDL model versus nearly all ceramides in the patient data. Since ACER3 is induced in both patients and the BDL model, B4GALT6 may account for these differences.

Response: Thank you for your positive assessment and insightful suggestions. Our data demonstrate that both *B4GALT6* and *ACER3* mRNA levels are upregulated in response to cholestasis and correlate with serum cholestatic liver injury severity markers (SCSMs) in patients (Figures 1B and 1C in our revised manuscript). However, by analyzing the correlation between dysregulated ceramide-related enzymes and SCSMs in patients with cholestatic liver injury, we found that *ACER3* mRNA levels exhibited the strongest positive correlations with a broader range of SCSMs compared to other enzymes, and the upregulation and correlation strength of *B4GALT6* were not as pronounced as those of *ACER3* (Figure 1C in our revised manuscript). This suggests that while *B4GALT6* may play a role in CER metabolism, *ACER3* appears to be the more significant regulator in our study. This supported our decision to focus primarily on ACER3 in this study. However, B4GALT6 is involved in the synthesis of lactosylceramides (LacCERs) and plays a role in inflammation regulation (PMID: 36982367, PMID: 25216636). We agree that *B4GALT6* may also contribute to lipid alterations and play a role in CLI pathogenesis. In our ongoing studies, we found that *B4galt6* was upregulated in the livers of human patients and mice with cholestasis (Figures C6A-C6D), and levels of LacCERs and glucosylceramides (GluCERs) were increased in the liver of female mice with cholestasis (Figure C6E). Administration of the B4galt6 inhibitor D-PDMP (PMID: 25216636) reduced inflammation and fibrosis in CLI without affecting Sult2a1 expression (Figures C6F-C6Q). In light of these

findings, we believe that *B4GALT6* is a promising target for future studies in the context of cholestasis and CER metabolism. We have revised the Results and Discussion sections to clarify our rationale for focusing on *ACER3* while encouraging future research on *B4GALT6* in CLI.

Figure C6

• Are there any flaws in the data analysis, interpretation and conclusions? Do these

prohibit publication or require revision?

Overall, the data was clear and well organized. One small complaint is the difficulty in evaluating the histology images as they were presented. The resolution is too low in the pdf, and I was unable to locate higher resolution figure files. This only had an impact on the ability to evaluate data for Fig. panels 1H, 2D, 2L, 3D, 3N, 4E, 4G, 4I, and 5B. The oil red images in particular were not very informative, it was difficult to distinguish background staining from staining of lipid droplets.

Response: Thank you for your positive evaluation of our data. We appreciate your feedback regarding the resolution of the histology images. We understand the importance of providing clear and detailed images, particularly for histological evaluation. We apologize for any difficulty this may have caused in assessing the figures. To address this concern, we have provided higher-resolution images of the relevant figures (panels 1G, 2D, 2L, 3D, 3N, 4I, 4K, 4L, and 5B) in the revised version of the manuscript. These images now have improved clarity and resolution, which should facilitate better evaluation of the histological findings.

Regarding the Oil Red O staining, we recognize that the previous images lacked a clear distinction between the lipid droplets and the background. We have updated the Oil Red O images with higher resolution and local magnification to more clearly highlight the lipid droplets. Additionally, we have revised the figures and figure legends to provide more detailed annotations and descriptions, guiding the reviewer to the regions of interest and helping to clarify the results.

We hope these revisions are sufficient to improve the visual clarity and make the histological data more accessible for evaluation. Thank you again for bringing this to our attention.

• Is the methodology sound? Does the work meet the expected standards in your

field?

The delivery of ceramide by IP with a carboxymethylcellulose vehicle is not typical. However, it was clearly effective for increasing hepatic d18:1/18:1 levels. Historically, it has been shown in the context of cell culture that ceramides with long acyl chains including d18:1/18:1 cer do not deliver well. Some studies, typically older ones, will employ a more soluble but artificial C6-ceramide. The C6-ceramide is rapidly remodeled leading to an increase in many sphingolipid species. In this historical context many researchers would avoid strategies that require exogenous delivery of ceramides. However, this strategy was clearly effective in this case. The point of these comments is that I feel that the approach would be more convincing to researchers in the sphingolipid field if the levels of all ceramide species were presented for this model instead of only d18:1/18:1 cer.

Response: We greatly appreciate your insightful comments on the methodology, particularly regarding the intraperitoneal (IP) delivery of ceramides using a carboxymethylcellulose (CMC) vehicle. We acknowledge that this approach is unconventional, especially given the historical challenges associated with delivering long-chain ceramides, such as ceramide(d18:1/18:1), in cell culture studies. As noted, despite these challenges, our strategy was effective in significantly increasing hepatic ceramide(d18:1/18:1) levels *in vivo*.

To provide a more comprehensive understanding and address your suggestion, we have now included a detailed lipidomic analysis covering all ceramide(d18:1) species in our revised manuscript. The newly added data demonstrated that the 10-day IP injection of ceramide(d18:1/18:1) not only markedly increased ceramide(d18:1/18:1) levels but also led to minimal increases in ceramide(d18:1/18:0) and ceramide(d18:1/20:1) in the liver tissues (Figures C10A and C10B). The increases of ceramide(d18:1/18:0) and ceramide(d18:1/20:1) likely reflects the highly dynamic nature of CER metabolism, wherein ceramide(d18:1/18:1) likely serves as a precursor in the biosynthesis of other

sphingolipids (SLs), such as sphingomyelins (SMs), glucosylceramides (GluCERs), and LacCERs, which may subsequently give rise to other ceramide species through reusing the breakdown products of these complex SLs (PMID: 29165427). Meanwhile, the degradation products of ceramide(d18:1/18:1), including SPH(d18:1) and fatty acid(18:1), may also be reutilized to generate other ceramides (PMID: 29165427). Despite the increase in other ceramide species, ceramide(d18:1/18:1) remained the most prominently elevated ceramide, highlighting the efficacy of the IP administration of ceramide(d18:1/18:1) in specifically increasing this ceramide in the liver, and supporting the model's relevance for studying the role of ceramide(d18:1/18:1) in CLI. Furthermore, we explored whether other ceramides generated during this process could also activate LXR β . Our immunoprecipitation experiments showed that ceramide(d18:1/18:1) was the most abundant ceramide bound to LXR β (Figure C10C). These findings collectively reinforce ceramide(d18:1/18:1) as a key mediator in regulating bile acid metabolism through its interaction with LXR β in cholestatic liver injury. We hope these newly included data will address your concerns and further validate our methodological approach. The additional lipidomics has been incorporated into Figure 5 and Supplementary Figures 8 and 12, and relevant updates have been made to the Results and Discussion Sections of the manuscript.

Figure C10

• Is there enough detail provided in the methods for the work to be reproduced?

The methods were sufficient and well-written.

Response: Thank you for your positive evaluation of our manuscript. We appreciate your acknowledgment that the methods are detailed and clear enough to ensure reproducibility.

Dear Reviewers,

We sincerely thank you for your thoughtful comments and the time you have dedicated to reviewing our work. We are pleased that our previous responses and revisions have addressed most of your concerns, and we deeply value your constructive feedback throughout this process. Your constructive insights have been invaluable in improving the quality and clarity of our manuscript.

The following are our point-by-point responses to the additional requests, questions, concerns, and comments. The corresponding modifications in our revised manuscript are highlighted in red text for clarity. Additionally, we have included supporting data labeled as Figure C in this rebuttal letter. These data have also been incorporated into the revised manuscript where relevant.

Reviewer #1 (Remarks to the Author):

Overall Review:

The authors have provided sincere and thorough responses to the reviewers' points, and the additional experiments are well-executed. In the previous review, we raised 9 Major Comments and 4 Minor Comments. Most of the Major Comments and all Minor Comments have been adequately addressed. However, I have additional comments regarding the authors' responses to some of the Major Comments. Addressing these issues would further enhance the quality of the manuscript.

Major Comments:

#1:

Response to Authors: Thank you for the detailed response. I have no further comments.

Response to Reviewer: We sincerely thank you for your time and effort in reviewing our manuscript. We greatly appreciate your positive feedback and are glad that our responses and revisions have addressed your concerns. Thank you again for your valuable input throughout the review process.

#2:

Response to Authors: Thank you for the detailed responses and the additional data.

Response to Reviewer: We are glad that our responses and revisions have addressed your concerns.

• **The authors state, “in ACER3-knockdown HepG2 cells, LXR α knockdown did not significantly reduce SULT2A1 expression” (Supplementary Figure 12I in the revised manuscript). However, the data in this figure appear to show a significant difference between shACER3-treated HepG2 cells treated with siCON and siLXR α . Could there be an error in the statistical analysis, or is there a specific rationale for interpreting these results as non-significant? Clarification on this point would be appreciated.**

Response to Reviewer: We appreciate the reviewer for raising this important point regarding the interpretation of the data of *SULT2A1* expression levels in *ACER3*-knockdown HepG2 cells. Upon revisiting our analysis, we confirm that *LXR α* knockdown (siLXR α) in *ACER3*-knockdown HepG2 cells (shACER3) resulted in a modest reduction in the mRNA levels of *SULT2A1* compared to the control siRNA (siCON) (Figure S13I in the revised manuscript). However, the magnitude of this reduction was minimal and did not translate into a statistically significant decrease in protein levels ($p > 0.05$) (Figures C1A and C1B). This discrepancy between mRNA and protein levels is likely due to the complex regulation of protein expression, where mRNA levels do not always correlate directly with protein levels (PMID: 28748932). Factors such as translation efficiency, protein stability, and post-translational modifications can influence protein expression and may contribute to this discrepancy (PMID: 32709985). Specifically, the modest decrease in the mRNA levels of *SULT2A1* may not have been sufficient to induce a corresponding reduction in its protein levels, particularly in *ACER3*-knockdown HepG2 cells, which inherently express high levels of *SULT2A1*. Consequently, we interpret the observed reduction in *SULT2A1* protein

levels in *ACER3*-knockdown HepG2 cells following *LXRα* knockdown as minimal and not statistically significant. In contrast, *LXRβ* knockdown in *ACER3*-knockdown HepG2 cells induced a more substantial and statistically significant reduction in both mRNA and protein levels of *SULT2A1* (Figure S13I in the revised manuscript and Figures C1B), highlighting the dominant role of *LXRβ* in mediating the regulatory effects of *ACER3* knockdown on *SULT2A1* expression. We hope this explanation clarifies our interpretation and addresses your concerns. We have revised the Results section to include this updated interpretation, and the statistical analysis of *SULT2A1* protein levels has been added to Supplementary Figure 13.

Figure C1

• **Additionally, while the authors explain that *LXRβ* is induced under *ACER3* knockdown conditions, the underlying mechanism of this upregulation remains unclear. Could the authors elaborate on how *ACER3* knockdown leads to increased *LXRβ* expression? For example, is this mediated through transcriptional regulation, post-transcriptional stabilization, or other pathways? Please discuss.**

Response to Reviewer: Thank you for raising this important point. As ligand-dependent nuclear receptors, LXRs are known to regulate their own expression in response to changes in ligand levels (PMID: 17189208 and 37890481). Our study demonstrated that CER(d18:1/18:1) serves as an agonist of *LXRβ*. However, the precise mechanisms by which *ACER3* and CER(d18:1/18:1) modulate *LXRβ*

expression and activity remain to be fully elucidated. Notably, LXR ligands have been shown to suppress the ubiquitination and degradation of LXRs or directly stabilize LXRs (PMID: 31799433 and 19164445). Thus, CER(d18:1/18:1) may confer ligand-mediated post-transcriptional stabilization to increase LXR β protein levels. Furthermore, previous studies have identified LXR response elements in the promoter region of the LXR genes, where the LXR/RXR heterodimer can bind and activate LXR expression, thereby forming a positive feedback loop (PMID: 11546778 and 15292368). Interestingly, our findings demonstrated that ACER3 knockout also led to the upregulation of RXR α protein levels (Figure S6J in the revised manuscript), suggesting that targeting the ACER3-CER(d18:1/18:1) metabolic axis may facilitate the formation of the LXR β /RXR α heterodimer, which in turn upregulates mRNA expression of LXR β via a positive feedback mechanism. As the reviewer correctly points out, the precise regulatory mechanisms underlying LXR β expression remain incompletely understood. Future research will be necessary to further elucidate how ACER3 and CER(d18:1/18:1) modulate LXR β expression and activity through transcriptional and post-transcriptional pathways. We have added this discussion to the revised manuscript.

#3:

Response to Authors: Thank you for the detailed response. The analysis of sex differences in each health condition (non-CLI and CLI) is thorough and well-presented. However, it would be beneficial to analyze differences within each sex according to health condition (e.g., “non-CLI vs. CLI in males” and “non-CLI vs. CLI in females”) and to discuss the results in this context.

Response to Reviewer: Thank you for the suggestion. We have conducted an analysis of hepatic ceramides (CERs) and their metabolizing enzymes in human populations with different sex backgrounds, considering both non-CLI and CLI conditions. Subgroup analyses comparing CLI and non-CLI groups revealed that cholestasis significantly upregulated the mRNA levels of *ACER3*, *ASAHL1*, *GLA*, *B4GALT6*, and

SGMS2 in male patients, while in female patients, it significantly increased the mRNA levels of *ACER3* and *DEGS2*. These results consolidated that the upregulation of hepatic *ACER3* by cholestasis is not sex-specific. Additionally, similar trends in the dysregulation of CER-metabolizing enzymes were observed across sexes: *DEGS2* showed a trend of upregulation in male patients, while *GLA*, *B4GALT6*, and *SGMS2* exhibited similar trends in female patients (Figure C2A). Notably, our original analyses comparing males and females within the same condition using both *in silico* GTEx Portal datasets and the collected liver tissues revealed no significant sex-specific differences in the mRNA levels of CER-related enzymes in either healthy or cholestatic liver tissues (Figures 7I and 7J in the revised manuscript). This apparent discrepancy may arise from differences in the analytical focus: the subgroup analysis comparing CLI versus non-CLI conditions aimed to investigate sex-specific differences in cholestasis-induced changes in certain CER-metabolizing enzymes, while the broader dataset analysis comparing males and females within the same health condition focused on identifying inherent sex-specific differences in the expression of CER-metabolizing enzymes. In terms of metabolites, comparisons of hepatic CER levels between CLI and non-CLI groups showed that cholestasis significantly increased the most CER species in both male and female patients (Figures C2B and C2C). This observation aligns with our original comparisons of hepatic CER levels between sexes within the same condition, which demonstrated no significant differences in most CER(d18:1) species between males and females in either non-CLI or CLI livers (Figures 7K and 7L in the revised manuscript). Collectively, these findings suggest that the overall increase in hepatic CERs, including CER(d18:1/18:1), induced by cholestasis is unlikely to be sex-specific. Similarly, cholestasis significantly decreased SPH levels in both male and female patients without affecting S1P levels (Figure C2D), and SPH and S1P levels showed no significant differences between sexes in either non-CLI or CLI livers (Figure C2E). In conclusion, these findings suggest that while cholestasis-induced changes in certain CER-metabolizing enzymes may involve subtle sex-specific regulation, the overall patterns of CER metabolism and activation of CER production are unlikely to be strongly influenced by sex. For instance, although *DEGS2* and *GLA* were found to

be differentially upregulated by cholestasis between sexes, both enzymes are known to be involved in CER generation (PMID: 34363020 and 8200356), which may contribute to the consistent increase in hepatic CER levels regardless of sex. However, we acknowledge the limitations of our sample size and emphasize the need for larger cohort studies to validate these observations and provide more robust insights into potential sex-specific differences in CER metabolism, particularly regarding specific CER-metabolizing enzymes. The new data and analyses have been incorporated into Supplementary Figure 10, and the Results and Discussion sections have been updated accordingly.

Figure C2

#4:

Response to Authors: Thank you for the detailed response. In the Western blot results (Figure C4-B), the expression level of *Sult2a1* between the control and knockdown groups does not appear to differ significantly. It would be beneficial to quantify the Western blot results to provide clearer evidence supporting your conclusion.

Response to Reviewer: Thank you for the valuable comment. In response, we have quantified and analyzed the Western blot results for *Sult2a1* expression in both the control and knockdown groups (Figure C3). The quantification results of the originally submitted Western blot are provided in Figures C3A and C3B (n = 3), and the combined quantification results with another three pairs of mice are provided in Figures C3C and C3D (n = 6). The updated analysis reveals a significant difference in *Sult2a1* expression between the control and knockdown groups. We hope these clarifications address your concerns. The manuscript has been updated to include the statistical analysis, which can be found in Supplementary Figure 6.

#5:

Response to Authors: Thank you for the detailed response. I have no further comments.

Response to Reviewer: We greatly appreciate your positive feedback and are glad that our responses and revisions have addressed your concerns.

#6:

Response to Authors: Thank you for the detailed response. I have no further comments.

Response to Reviewer: We are pleased that our responses have adequately addressed your comments and appreciate your valuable input in refining our manuscript.

#7:

Response to Authors: Thank you for the detailed response. I have no further comments.

Response to Reviewer: We are pleased that our responses have adequately addressed your comments and appreciate your valuable input in refining our manuscript.

#8:

Response to Authors: Thank you for the detailed response. I have no further comments.

Response to Reviewer: We greatly appreciate your positive feedback and are glad that our responses and revisions have addressed your concerns.

#9:

Response to Authors: Thank you for the detailed response. I have no further comments.

Response to Reviewer: We greatly appreciate your positive feedback and are glad that our responses and revisions have addressed your concerns.

Minor Comments:

Thank you for the detailed response. I have no further comments.

Response to Reviewer: We are deeply grateful to the reviewers for their insightful suggestions and thorough evaluation of our work. We are delighted that our revisions and responses have resolved all minor concerns, and we appreciate the time and effort you have dedicated to improving the quality of our study.

Reviewer #2 (Remarks to the Author):

- Does the work support the conclusions and claims, or is additional evidence needed?

The most consequential conclusions are very well-supported by the data. Specifically, the role of Acer3 and ceramides in mediating BA sulfation, the effect on expression of LXRalpha and its targets, and the impact on the severity of BDL-induced disease. I do not see any gaps that would prevent publication, however there were a few questions I felt were unanswered that may represent opportunities to improve the manuscript.

Revised response: I feel that my initial comment was adequately addressed. I am satisfied with the revised Results and Discussion section.

Response to Reviewer: We sincerely thank you for your feedback and are pleased to hear that our revisions have adequately addressed your initial comment.

- Are there any flaws in the data analysis, interpretation and conclusions? Do these prohibit publication or require revision? Overall, the data was clear and well organized. One small complaint is the difficulty in evaluating the histology images as they were presented. The resolution is too low in the pdf, and I was unable to locate higher resolution figure files. This only had an impact on the ability to evaluate data for Fig. panels 1H, 2D, 2L, 3D, 3N, 4E, 4G, 4I, and 5B. The oil red images in particular were not very informative, it was difficult to distinguish background staining from staining of lipid droplets.

Revised response: I am satisfied my initial comment was adequately addressed with the availability of higher quality images and the revised figure legends.

Response to Reviewer: We sincerely thank you for your positive feedback and are delighted to hear that our revisions, including the availability of higher-quality images and the updated figure legends, have satisfactorily addressed your initial comment.

- Is the methodology sound? Does the work meet the expected standards in your field? The delivery of ceramide by IP with a carboxymethylcellulose vehicle is not typical. However, it was clearly effective for increasing hepatic d18.1/18.1 levels. Historically, it has been shown in the context of cell culture that ceramides with long acyl chains

including d18.1/18.1 cer do not deliver well. Some studies, typically older ones, will employ a more soluble but artificial C6-ceramide. The C6-ceramide is rapidly remodeled leading to an increase in many sphingolipid species. In this historical context many researchers would avoid strategies that require exogenous delivery of ceramides. However, this strategy was clearly effective in this case. The point of these comments is that I feel that the approach would be more convincing to researchers in the sphingolipid field if the levels of all ceramide species were presented for this model instead of only d18.1/18.1 cer.

Revised response: I appreciate the inclusion of more detailed lipidomics data, and I am sure these will be appreciated by the readers as well. I am more than satisfied that the newly incorporated data addresses my initial comments.

Response to Reviewer: We sincerely thank you for your positive feedback. We are delighted to hear that the newly incorporated lipidomics data have not only addressed your initial comments but will also provide additional value for the readers. Your thoughtful suggestions have greatly contributed to enhancing the quality and impact of our manuscript.